# CLOSED-LOOP TRANSCRIPTION VIA CONVOLUTIONAL SPARSE CODING

## ABSTRACT

Autoencoding has been a popular and effective framework for learning generative models for images, with much empirical success. Autoencoders often use generic deep networks as the encoder and decoder, which are difficult to interpret, and the learned representations lack clear structure. In this work, we replace the encoder and decoder with standard *convolutional sparse coding* and decoding layers, obtained from unrolling an optimization algorithm for solving a (convexified) sparse coding program. Furthermore, to avoid computational difficulties in minimizing distributional distance between the real and generated images, we utilize the recent *closed-loop transcription* (CTRL) framework that maximizes the rate reduction of the learned sparse representations. We show that such a simple framework demonstrates surprisingly competitive performance on large datasets, such as ImageNet-1K, compared to existing autoencoding and generative methods under fair conditions. Even with simpler networks and less computational resources, our method demonstrates high visual quality in regenerated images with good sample-wise consistency. More surprisingly, the learned autoencoder generalizes to unseen datasets. Our method enjoys several side benefits, including more structured and interpretable representations, more stable convergence, scalability to large datasets – indeed, our method is the *first* sparse coding generative method to scale up to ImageNet – and trainability with smaller batch sizes.

## 1 INTRODUCTION

In recent years, deep networks have been widely used to learn generative models for real images, via popular methods such as generative adversarial networks (GAN) (Goodfellow et al., 2020), variational autoencoders (VAE) (Kingma & Welling, 2013), and score-based diffusion models (Hyvärinen, 2005; Sohl-Dickstein et al., 2015; Ho et al., 2020). Despite tremendous empirical successes and progress, these methods typically use empirically designed, or generic, deep networks for the encoder and decoder (or discriminator in the case of GAN). As a result, how each layer generates or transforms imagery data is not interpretable, and the internal structures of the learned representations remain largely unrevealed. Further, the true layer-by-layer interactions between the encoder and decoder remain largely unknown. These problems often make the network design for such methods uncertain, training of such generative models expensive, the resulting representations hidden, and the images difficult to be conditionally generated. The recently proposed closed-loop transcription (CTRL) (Dai et al., 2022b) framework aims to learn autoencoding models with more structured representations by maximizing the information gain, say in terms of the coding rate reduction (Ma et al., 2007; Yu et al., 2020) of the learned features. Nevertheless, like the aforementioned generative methods, CTRL uses two separate generic encoding and decoding networks which limit the true potential of such a framework, as we will discuss later.

On the other side of the coin, in image processing and computer vision, it has long been believed and advocated that sparse convolution or deconvolution is a conceptually simple generative model for natural images (Monga et al., 2021). That is, natural images at different spatial scales can be explicitly modeled as being generated from a sparse superposition of a number of atoms/motifs, known as a (convolution) dictionary. There has been a long history in image processing of using such models for applications such as image denoising, restoration, and superresolution (Elad & Aharon, 2006; Elad, 2010; Yang et al., 2010). Some recent literature has also attempted to use sparse convolution as building blocks for designing more intepretable deep networks (Sulam et al., 2018). One conceptual benefit of such a model is that the encoding and decoding can be interpreted as mutually invertible

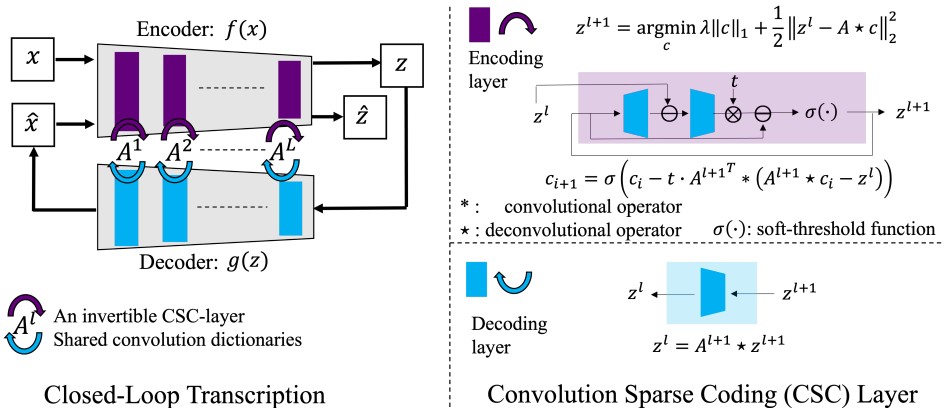

Figure 1: **Left:** A CTRL architecture with convolutional sparse coding layers in which the encoder and decoder share the same convolution dictionaries. **Right:** the encoder of each convolutional sparse coding layer is simply the unrolled optimization for convolutional sparse coding (e.g. ISTA/FISTA).

(sparse) convolution and deconvolution processes, respectively, as illustrated in Figure 1 right. At each layer, instead of using two separate convolution networks with independent parameters, the encoding and decoding processes share the same learned convolution dictionary. This has been the case for most aforementioned generative or autoencoding methods. Despite their simplicity and clarity, most sparse convolution based deep models are limited to tasks like image denoising (Mohan et al., 2019) or image restoration (Lecouat et al., 2020). Their empirical performance on image generation or autoencoding tasks has not yet been shown as competitive as the above mentioned methods (Aberdam et al., 2020), in terms of either image quality or scalability to large datasets.

Hence, in this paper, we try to investigate and resolve the following question: *can we use convolutional sparse coding layers to build deep autoencoding models whose performance can compete with, or even surpass, that of tried-and-tested deep networks?* Although earlier attempts to incorporate such layers within the GAN and VAE frameworks have not resulted in competitive performance, the invertible convolutional sparse coding layers are naturally compatible with the objectives of the recent closed-loop transcription CTRL framework (Dai et al., 2022b), see Figure 1 left. CTRL utilizes a self-critiquing sequential maximin game (Pai et al., 2022) between the encoder and decoder to optimize the coding rate reduction of the learned internal (sparse) representations. Such a self-critiquing mechanism can effectively enforce the learned convolution dictionaries, now shared by the encoder and decoder at each layer, to strike a better tradeoff between coding compactness and generation quality. The closed-loop framework also avoids any computational caveats associated with frameworks such as GAN and VAE for evaluating (or approximating) distribution distances in the pixel space.

As we will show in this paper, by simply using invertible convolutional sparse coding layers[1] within the CTRL framework, the performance of CTRL can be significantly improved compared to using two separate networks for the encoder and decoder. For instance, as observed in Dai et al. (2022b), the autoencoding learned by CTRL successfully aligns the distributions of the real and generated images in the feature space, but fails to achieve good sample-wise autoencoding. In this work, we show that CTRL can now achieve precise sample-wise alignment with the convolutional sparse coding layers. In addition, we show that deep networks constructed purely with convolutional sparse coding layers yield superior practical performance for image generation, with fewer model parameters and less computational cost. Our work provides compelling empirical evidence which suggests that a multi-stage sparse (de)convolution has the potential to serve as a backbone operator for image generations.

To be more specific, we will see through extensive experiments that the proposed simple closed-loop transcription framework with transparent and interpretable sparse convolution coding layers enjoy the following benefits:

1. *Good performance on large datasets.* Compared to previous sparse coding based generative or autoencoding methods, our method scales well to large datasets such as ImageNet-1k,

---

[1]Based on the implementation suggested by Zeiler et al. (2010).

with a comparable performance to the common generative or autoencoding methods based on GAN or VAE, under fair experimental comparisons.

2. *Better sample-wise alignment and dataset generalizability.* The learned autoencoder achieves good sample-wise consistency despite only optimizing alignment between distributions. We also show the generalizability of the CSC based autoencoder – an autoencoder trained on CIFAR-10 can be applied to reconstruct CIFAR-100.

3. *More structured representations.* The learned feature representations for each class of images tend to have sparse low-dimensional linear structure that is amenable for conditional image generation.

4. *Higher efficiency and stability.* Our method can achieve comparable or better performance than other autoencoding methods with smaller networks, smaller training batch sizes, and faster convergence. The autoencoder learned is more stable to noise than autoencoding models based on generic networks.

## 2 RELATED WORK

**Sparse Dictionary Learning.** Inspired by neuroscience studies (Olshausen & Field, 1996; 1997), sparse coding or sparse dictionary learning (SDL) has a long history and numerous applications in modeling high-dimensional data, especially images (Argyriou et al., 2008; Elad & Aharon, 2006; Wright et al., 2008; Yang et al., 2010; Mairal et al., 2014; Wright & Ma, 2022). Specifically, given a dataset $\{y_i\}_{i=1}^n$, SDL considers the problem of learning an dictionary $A$ such that $y_i$ has sparse representations $y_i \approx Ax_i, \forall i \in [n]$ with $x_i$ sparse. To understand the theoretical tractability of SDL, several lines of works based on $\ell^1$-norm minimization (Spielman et al., 2012; Geng & Wright, 2014; Sun et al., 2015; Bai et al., 2018; Gilboa et al., 2018), $\ell^p$-norm maximization (for $p \geq 3$) (Zhai et al., 2020; 2019; Shen et al., 2020; Qu et al., 2019), and sum-of-squares methods have been proposed (Barak et al., 2015; Ma et al., 2016; Schramm & Steurer, 2017). Inspired by the empirical success of SDL on tasks such as face recognition and image denoising Wright et al. (2008); Mairal et al. (2008; 2009; 2012; 2014), our convolutional sparse coding-based networks seek to learn a sparse representation from input images and use the learned sparse representation for image generation or autoencoding purposes. In particular, we adopt the classic $\ell^1$-minimization framework for learning sparse latent representations, as the $\ell^1$ norm can be viewed as a convex surrogate of the $\ell^0$ norm, which is a direct indicator of sparsity (Wright & Ma, 2022).

**Convolutional Sparse Coding Layer.** The idea of using sparse coding for network architectures can be traced back to the work of unrolling sparse coding algorithms such as ISTA to learn the sparsifying dictionary (Gregor & LeCun, 2010; Tolooshams & Ba, 2021) and some deconvolutional networks(Zeiler et al., 2010; 2011; Papyan et al., 2017). Several recent works have explored deep networks with convolutional sparse coding layers for image denoising, image restoration, and (network normalization for) image classification (Sreter & Giryes, 2018; Mohan et al., 2019; Lecouat et al., 2020; Liu et al., 2021). The validity of these networks has mostly been demonstrated on datasets with small or moderate scales, especially for tasks such as image classification or generation. Recently the SD-Net (Dai et al., 2022a) successfully demonstrated that convolutional sparse coding inspired networks can perform well on image classification tasks on large image datasets such as ImageNet-1K. Encouraged by such successes, our work seeks to further validate the capability of the family of convolutional sparse coding-based networks for the more challenging image generation and autoencoding tasks on large-scale datasets.

**Sparse Modeling for Generative Models.** We are not the first to consider incorporating sparse modeling to facilitate generative tasks. To our knowledge, most existing approaches focus around using sparsity to improve GANs. For example, Mahdizadehaghdam et al. (2019) exploits patch-based sparsity and takes in a pre-trained dictionary to assembling generated patches. Ganz & Elad (2021) explores convolutional sparse coding in generative adversarial networks, arguing that the generator is a manifestation of the convolutional sparse coding and its multi-layered version synthesis process. Both methods have shown that using sparsity-inspired networks improves the image quality of GANs. However, these two works either use a pretrained dictionary or limit to smaller scales of data, such as the CIFAR-10 dataset. Aberdam et al. (2020) uses sparse representation theory to study the inverse problem in image generation. They developed a two-layer inversion pursuit algorithm for training generative models for imagery data. On datasets like MNIST, Aberdam et al. (2020) shows that generative models can be inverted. Nonetheless, most sparse-coding inspired generative frameworks

have only been shown to work on smaller datasets like MNIST and CIFAR-10. In this work, we demonstrate that by incorporating convolutional sparse coding into a proper generative framework, namely CTRL, the convolutional sparse coding-based networks demonstrate good performance on large datasets, and also have several good benefits (refer to section 4.4 and 4.3).

## 3 OUR METHOD

Our goal is to learn an autoencoder from large image datasets that can achieve both distribution-wise and sample-wise autoencoding with high image quality. Our method will be based on a classic generative model for natural images: a multi-layer sparse (de)convolution model. The autoencoding will be established through learning the (de)convolution dictionaries at all layers based on the recent closed-loop transcription framework, as it naturally optimizes coding rates of the sparse representations sought by the convolutional sparse coding.

**Autoencoding and Its Caveats.** In classic autoencoding problems, we consider a random vector $x \in \mathbb{R}^D$ in a high-dimensional space whose distribution is typically supported on a low-dimensional submanifold $\mathcal{M}$. We seek a continuous encoding mapping $f(\cdot, \theta)$, say parameterized by $\theta$, that maps $x \in \mathbb{R}^D$ to a compact feature vector $z = f(x)$ in a much lower-dimensional space $\mathbb{R}^d$. In addition, we also seek an (inverse) decoding mapping $g(\cdot, \eta)$, parameterized by $\eta$, that maps the feature $z$ back to the original data space $\mathbb{R}^D$:

$$f(\cdot, \theta) : x \mapsto z \in \mathbb{R}^d; \quad g(\cdot, \eta) : z \mapsto \hat{x} \in \mathbb{R}^D \tag{1}$$

in such a way that $x$ and $\hat{x} = g(f(x))$ are "close," i.e., some distance measure $\mathcal{D}(x, \hat{x})$ is small.

In practice, we only have a set of $n$ samples $X = [x^1, \ldots, x^n]$ of $x$. Let $Z = f(X, \theta) \doteq [z^1, \ldots, z^n] \subset \mathbb{R}^{d \times n}$ with $z^i = f(x^i, \theta) \in \mathbb{R}^d$ be the set of corresponding features. Similarly let $\hat{X} \doteq g(Z, \eta)$ be the decoded data from the features. The overall autoencoding process can be illustrated by the following diagram:

$$X \xrightarrow{f(x, \theta)} Z \xrightarrow{g(z, \eta)} \hat{X}. \tag{2}$$

In general, we wish that $\hat{X}$ is close to $X$ based on some distance measure $\mathcal{D}(X, \hat{X})$. In particular, we often wish that for each sample $x^i$, the distance between $x^i$ and $\hat{x}^i$ is small.

However, the nature of the distribution of images is typically unknown. Historically, this has caused two fundamental difficulties associated with obtaining a good autoencoding (or generative model) for imagery data. First, it is normally very difficult to find a principled, computable, and well-defined distance measure between the distributions of two image datasets, say $X$ and $\hat{X}$. This is the fundamental reason why in GAN (Goodfellow et al., 2020), a discriminator was introduced to replace the conceptual role of such a distance; and in VAE (Kingma & Welling, 2013), variational bounds were introduced to approximate such a distance. Second, most methods do not start with a clear generative model for images and instead adopt generic convolution neural networks for the encoder and decoder (or discriminator). Such networks do not have clear mathematical interpretations; also, it is difficult to enforce sample-wise invertibility of the networks (Dai et al., 2022b).

Below, we show how both difficulties can be explicitly and effectively addressed in our approach. Our approach starts from a simple and clear model of image generation.

### 3.1 CONVOLUTIONAL SPARSE CODING

**A Generative Model for Images as Sparse Deconvolution.** We may consider an image $x$, or its representation at any given stage of a deep network, as a multi-dimensional signal $x \in \mathbb{R}^{M \times H \times W}$ where $H, W$ are spatial dimensions and $M$ is the number of channels. We assume the image $x$ is generated by a multi-channel sparse code $z \in \mathbb{R}^{C \times H \times W}$ deconvolving with a multi-dimensional kernel $A \in \mathbb{R}^{M \times C \times k \times k}$, which is referred to as a *convolution dictionary*. Here $C$ is the number of channels for $z$ and the convolution kernel $A$. To be more precise, we denote $z$ as $z \doteq (\zeta_1, \ldots, \zeta_C)$ where each $\zeta_c \in \mathbb{R}^{H \times W}$ is a 2D array (presumably sparse), and denote the kernel $A$ as

$$A \doteq \begin{pmatrix} \alpha_{11} & \alpha_{12} & \alpha_{13} & \ldots & \alpha_{1C} \\ \alpha_{21} & \alpha_{22} & \alpha_{23} & \ldots & \alpha_{2C} \\ \vdots & \vdots & \vdots & \ddots & \vdots \\ \alpha_{M1} & \alpha_{M2} & \alpha_{M3} & \ldots & \alpha_{MC} \end{pmatrix} \in \mathbb{R}^{M \times C \times k \times k}, \tag{3}$$

where each $\boldsymbol{\alpha}_{ij} \in \mathbb{R}^{k \times k}$ is a 2D motif of size $k \times k$. Then, for each layer of the generator, also called the decoder, $g(\boldsymbol{z}, \eta)$, its output signal $\boldsymbol{x}$ is generated via the following operator $\mathcal{A}(\cdot)$ defined by deconvolving the dictionary $\boldsymbol{A}$ with the sparse code $\boldsymbol{z}$:

$$\boldsymbol{x} = \mathcal{A}(\boldsymbol{z}) + \boldsymbol{n} \doteq \sum_{c=1}^{C} \left(\boldsymbol{\alpha}_{1c} \star \boldsymbol{\zeta}_c, \dots, \boldsymbol{\alpha}_{Mc} \star \boldsymbol{\zeta}_c\right) + \boldsymbol{n} \quad \in \mathbb{R}^{M \times H \times W}. \tag{4}$$

where $\boldsymbol{n}$ is some small isotropic Gaussian noise. For convenience, we use "$*$" and "$\star$" to denote the convolution and deconvolution operators, respectively, between any two 2D signals $(\boldsymbol{\alpha}, \boldsymbol{\zeta})$:

$$(\boldsymbol{\alpha} * \boldsymbol{\zeta})[i,j] \doteq \sum_p \sum_q \boldsymbol{\zeta}[i-p, j-q] \cdot \boldsymbol{\alpha}[p,q], \quad (\boldsymbol{\alpha} \star \boldsymbol{\zeta})[i,j] \doteq \sum_p \sum_q \boldsymbol{\zeta}[i+p, j+q] \cdot \boldsymbol{\alpha}[p,q]. \tag{5}$$

Hence, the decoder $g(\boldsymbol{x}, \eta)$ is a concatenation of multiple such sparse deconvolution layers and the parameters $\eta$ are the collection of convolution dictionaries $\boldsymbol{A}$'s (to be learned). Only batch normalization and ReLU are added between consecutive layers to normalize the overall scale and to ensure positive pixel values of the generated images. Details can be found in the Appendix A.

**An Encoding Layer as Convolutional Sparse Coding.** Now, given a multi-dimensional input $\boldsymbol{x} \in \mathbb{R}^{M \times H \times W}$ sparsely generated from a (learned) convolution dictionary $\boldsymbol{A}$, the function of "each layer" of the encoder $f(\boldsymbol{x}, \theta)$ is to find the optimal $\boldsymbol{z}_* \in \mathbb{R}^{C \times H \times W}$ from solving the inverse problem from equation 4. Under the above sparse generative model, according to (Wright & Ma, 2022), we can seek the optimal sparse solution $\boldsymbol{z}$ by solving the following Lasso type optimization problem:

$$\boldsymbol{z}_* = \underset{\boldsymbol{z}}{\operatorname{argmin}} \left\{ \lambda \|\boldsymbol{z}\|_1 + \frac{1}{2} \|\boldsymbol{x} - \mathcal{A}(\boldsymbol{z})\|_2^2 \right\} \quad \in \mathbb{R}^{C \times H \times W}. \tag{6}$$

We refer to such an implicit layer defined by equation 6 as a convolutional sparse coding layer. The difference between $\boldsymbol{x}$ and $\mathcal{A}(\boldsymbol{z})$ is penalized by the $\ell_2$-norm of $\boldsymbol{x} - \mathcal{A}(\boldsymbol{z})$ flattened into a vector.

The optimal solution of $\boldsymbol{z}$ given $\mathcal{A}$ will be a close reconstruction of $\boldsymbol{x}$. Sparsity is controlled by the entry-wise $\ell_1$-norm of $\boldsymbol{z}$ in the objective. $\lambda$ controls the level of desired sparsity. In this paper, we adopt the the fast iterative shrinkage thresholding algorithm (FISTA) (Beck & Teboulle, 2009) for the forward propagation. The basic iterative operation is illustrated in Figure 1. A natural benefit of the FISTA algorithm is that it leads to a network architecture that is constructed from an unrolled optimization algorithm, for which backward propagation can be carried out by auto-differentiation.

Hence, the encoder $f(\boldsymbol{x}, \theta)$ is a concatenation of such convolutional sparse coding layers (again, with batch normalization and ReLU). Recently, the work of (Dai et al., 2022a) has shown that such a convolution sparse coding network demonstrates competitive performance against popular deep networks such as the ResNet in large-scale image classification tasks. Note that in the generative setting, the operators of each layer of the encoder $f$ are determined by the same collection of convolution dictionaries $\boldsymbol{A}$'s as the decoder $g$. Thus, in the autoencoding diagram 2, the parameters $\theta$ of the encoder $f(\boldsymbol{x}, \theta)$ and $\eta$ of the decoder $g(\boldsymbol{x}, \eta)$ are determined by the same dictionaries. As we will see, this coupling benefits the learned autoencoder, even besides interpretability.

## 3.2 CLOSED-LOOP TRANSCRIPTION (CTRL)

The above explicit generative model has resolved the issue regarding the structure of the the encoder and decoder for the autoencoding[2]: $\boldsymbol{X} \xrightarrow{f(\boldsymbol{x}, \theta)} \boldsymbol{Z} \xrightarrow{g(\boldsymbol{z}, \eta)} \hat{\boldsymbol{X}}$. It does not yet address another difficulty mentioned above about autoencoding: how should we measure the difference between $\boldsymbol{X}$ and the regenerated $\hat{\boldsymbol{X}} = g(f(\boldsymbol{X}, \theta), \eta)$? As we discussed earlier, it is difficult to identify the correct distance between (distributions of) images. Nevertheless, if we believe the images are sparsely generated and the sparse codes can be correctly identified through the above mappings, then it is natural to measure the distance in the learned (sparse) feature space.

The recently proposed *closed-loop transcription* (CTRL) framework proposed by Dai et al. (2022b) is designed for this purpose. The difference between $\boldsymbol{X}$ and $\hat{\boldsymbol{X}}$ can be measured through the distance between their corresponding features $\boldsymbol{Z}$ and $\hat{\boldsymbol{Z}} = f(\hat{\boldsymbol{X}}, \theta)$ mapped through the same encoder:

$$\boldsymbol{X} \xrightarrow{f(\boldsymbol{x}, \theta)} \boldsymbol{Z} \xrightarrow{g(\boldsymbol{z}, \eta)} \hat{\boldsymbol{X}} \xrightarrow{f(\boldsymbol{x}, \theta)} \hat{\boldsymbol{Z}}. \tag{7}$$

---

[2] Although the effectiveness of the choice remains to be verified.

Their distance can be measured by the so-called rate reduction (Ma et al., 2007; Yu et al., 2020): namely the difference between the rate distortion of the union of $\boldsymbol{Z}$ and $\hat{\boldsymbol{Z}}$ and the sum of their individual rate distortions:

$$\Delta R(\boldsymbol{Z}, \hat{\boldsymbol{Z}}) \doteq R(\boldsymbol{Z} \cup \hat{\boldsymbol{Z}}) - \frac{1}{2}\big(R(\boldsymbol{Z}) + R(\hat{\boldsymbol{Z}})\big). \tag{8}$$

where $R(\cdot)$ represents the rate distortion function of a distribution. In the case of $\boldsymbol{Z}$ being a Gaussian distribution and for any given allowable distortion $\epsilon > 0$, $R(\boldsymbol{Z})$ can be closedly approximated by $\frac{1}{2} \log \det \big(\boldsymbol{I} + \frac{d}{n\epsilon^2} \boldsymbol{Z}\boldsymbol{Z}^\top\big)$.[3] Such a $\Delta R$ gives a principled distance between subspace-like Gaussian ensembles, with the property that $\Delta R(\boldsymbol{Z}, \hat{\boldsymbol{Z}}) = 0$ iff $\text{Cov}(\boldsymbol{Z}) = \text{Cov}(\hat{\boldsymbol{Z}})$ (Ma et al., 2007).

As shown in (Dai et al., 2022b; Pai et al., 2022), one can provably learn a good autoencoding by allowing the encoder and decoder to play a sequential game: the encoder $f$ plays the role of discriminator to separate $\boldsymbol{Z}$ and $\hat{\boldsymbol{Z}}$ and $g$ plays as a generator to minimize the difference. This results in the following maxmin program:

$$\max_\theta \min_\eta \ \Delta R\big(\boldsymbol{Z}(\theta), \hat{\boldsymbol{Z}}(\theta, \eta)\big). \tag{9}$$

The program in equation 9 is somewhat limited because it only aims to align the dataset $\boldsymbol{X}$ and the regenerated $\hat{\boldsymbol{X}}$ at the distribution level. There is no guarantee that for each sample $\boldsymbol{x}^i$ would be close to the regenerated $\hat{\boldsymbol{x}}^i = g(f(\boldsymbol{x}^i, \theta), \eta)$. For example, Dai et al. (2022b) shows that an input image of a car can be decoded into a horse; the so obtained autoencoding is not sample-wise consistent.

A likely reason for this to happen is because two separate networks are used for the encoder and decoder and the rate reduction objective function only minimizes error between distributions, not individual samples. Now notice that for the new convolutional sparse coding layers, parameters of the encoder $f$ and decoder $g$ are determined by the same convolution dictionaries $\boldsymbol{A}$. Hence the above rate reduction objective in equation 9 becomes a function of $\boldsymbol{A}$:

$$\Delta R\big(\boldsymbol{Z}(\theta(\boldsymbol{A})), \hat{\boldsymbol{Z}}(\theta(\boldsymbol{A}), \eta(\boldsymbol{A}))\big). \tag{10}$$

We can use this as a cost function to guide us to learn dictionaries $\boldsymbol{A}$ which are discriminative for the inputs and able to represent them faithfully through closed-loop transcription. To this end, for each batch of new data samples, we take one ascent step and then one descent step. The first, maximizing, step promotes a discriminative sparse encoder using only the encoder gradient, and the second, minimizing, step promotes a consistent autoencoding by using the gradients of the entire closed loop.

$$\max_{\theta(\boldsymbol{A})} \Delta R \ \text{step}: \quad \boldsymbol{A}_{k+1} = \boldsymbol{A}_k + \lambda_{\max} \frac{\partial \Delta R}{\partial \theta} \cdot \frac{\partial \theta}{\partial \boldsymbol{A}} \Big|_{\boldsymbol{A}_k}, \tag{11}$$

$$\min_{\boldsymbol{A}} \Delta R \ \text{step}: \quad \boldsymbol{A}_{k+2} = \boldsymbol{A}_{k+1} - \lambda_{\min} \Big(\frac{\partial \Delta R}{\partial \eta} \cdot \frac{\partial \eta}{\partial \boldsymbol{A}} + \frac{\partial \Delta R}{\partial \theta} \cdot \frac{\partial \theta}{\partial \boldsymbol{A}}\Big) \Big|_{\boldsymbol{A}_{k+1}}. \tag{12}$$

Empirically, we find that in the step to minimize $\Delta R$, taking the gradient as the total derivative with respect to the dictionary $\boldsymbol{A}$, i.e., using the gradients through both $\theta$ and $\eta$, converges to better results than just using the gradient through $\eta$ – see the ablation studies of Appendix B. As we will see, by sharing convolution dictionaries in the encoder and decoder, the learned autoencoder can achieve good sample-wise consistency even though the rate reduction objective in equation 10 is meant to promote only distributional alignment.

## 4 EXPERIMENTS

We now evaluate the effectiveness of the proposed method. The main message we want to convey is that the convolutional sparse coding-based deep models can indeed scale up to large-scale datasets and regenerate high-quality images. Note that the purpose of our experiments is *not* to claim we can achieve state-of-the-art performance compared to all existing generative or autoencoding methods, including those that may have much larger model complexities and require arbitrary amounts of data and computational resources.[4] We compare our method with several representative categories of generative or autoencoding models, under fair[5] experimental conditions: for instance, since our

---

[3] We refer to (Dai et al., 2022b) for more general cases such as a mixture of Gaussians or subspaces.

[4] For example, we will not compare with methods that require very large models such as Big-GAN (Brock et al., 2018) or NSCN++ (Song et al., 2020).

[5] or less fair to ours.

| Method | CIFAR-10 | | STL-10 | | ImageNet | |
|---|---|---|---|---|---|---|
| | IS↑ | FID↓ | IS↑ | FID↓ | IS↑ | FID↓ |
| *GAN based methods* | | | | | | |
| DCGAN (Radford et al., 2015) | 6.6 | 35.3 | 7.8 | - | - | - |
| SNGAN (Miyato et al., 2018) | 7.4 | 29.3 | 9.1 | 40.1 | 7.3 | 48.7 |
| *VAE based methods* | | | | | | |
| VAE (Kingma & Welling, 2013) | 5.2 | 55.9 | - | - | - | - |
| NVAE (Vahdat & Kautz, 2020) | - | 50.8 | - | - | - | - |
| *Flow based methods* | | | | | | |
| GLOW (Kingma & Dhariwal, 2018) | - | 46.9 | - | - | - | - |
| Residual Flow (Chen et al., 2019) | - | 50.8 | - | - | - | - |
| *CTRL based methods* | | | | | | |
| CTRL (Dai et al., 2022b) | 8.1 | 19.6 | 8.4 | 38.6 | 7.7 | 46.9 |
| CSC-CTRL (ours) | 8.9 | 28.9 | 9.1 | 48.1 | 12.5 | 34.5 |

Table 1: Comparison on CIFAR-10, STL-10, and ImageNet-1K. The network architectures used in CSC-CTRL are 4-layers for CIFAR-10, 5-layers for STL-10 and ImageNet respectively which are much smaller than other compared methods.

method uses only two simple networks, we mainly compare with methods using two networks[6], e.g., one for encoder (or generator) and one for decoder (or discriminator).

**Datasets and Experiment Setting.** We test the performance of our method on CIFAR-10 (Krizhevsky et al., 2009), STL-10 (Coates et al., 2011) and ImageNet-1k (Deng et al., 2009) datasets. Implementation details for CIFAR-10, STL-10 and ImageNet-1k are in Appendix A.1.

## 4.1 PERFORMANCE ON IMAGE AUTOENCODING

We adopt the standard FID (Heusel et al., 2017) and Inception Score (IS) (Salimans et al., 2016) to evaluate autoencoding quality. We compare our method to the most representative methods from the following categories: GAN, VAE, flow-based, and CTRL, under the same experimental conditions – except that our method typically uses simpler and smaller models.

On medium-size datasets such as CIFAR-10, we observe in Table 1 that, in terms of these metrics, our method achieves comparable or better performance compared to typical GAN, flow-based and VAE methods, and better IS than CTRL and VAE-based methods, which conceptually are the closest to our method. Comparing to CTRL, Fig 2 showcases the different reconstructed image between CTRL and CSC-CTRL. It is clear that CSC-CTRL not only enjoys better visual quality, but also achieves much better sample-wise alignment. Visually, Figure 3 further shows sample-wise alignment between input $X$ and reconstructed $\hat{X}$ despite our method not enforcing sample-wise constraints or loss functions!

On larger-scale datasets such as ImageNet-1k, Table 1 shows that we outperform many existing methods in Inception Score. Figure 3 shows that the decoded $\hat{X}$ looks almost identical to the original $X$, even in tiny details. All of the images displayed are randomly chosen without cherry-picking. Due to page limitations, we place more results on ImageNet and STL-10 in Appendix C.

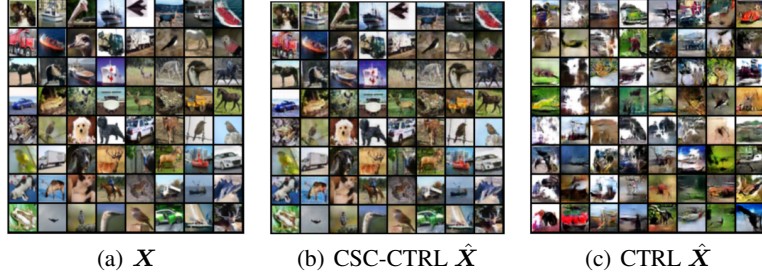

(a) $X$       (b) CSC-CTRL $\hat{X}$       (c) CTRL $\hat{X}$

Figure 2: Visualizing the auto-encoding property of the learned CSC-CTRL ($\hat{X} = g \circ f(X)$) comparing to CTRL on CIFAR-10. (Images are randomly chosen.)

---

[6]Hence, we will not compare with methods that require multiple networks for additional discriminators such as the VAE-GAN (Parmar et al., 2021) and the Style-GAN (Karras et al., 2020).

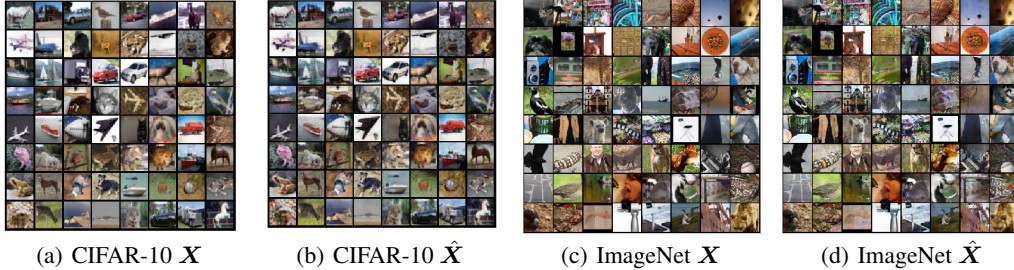

| (a) CIFAR-10 $\boldsymbol{X}$ | (b) CIFAR-10 $\hat{\boldsymbol{X}}$ | (c) ImageNet $\boldsymbol{X}$ | (d) ImageNet $\hat{\boldsymbol{X}}$ |

Figure 3: Visualizing the auto-encoding property of the learned CSC-CTRL ($\hat{\boldsymbol{X}} = g \circ f(\boldsymbol{X})$) on CIFAR-10 and ImageNet. (Images are randomly chosen.)

## 4.2 STRUCTURES OF LEARNED REPRESENTATIONS

To evaluate the structural properties of the learned feature space, we visualize the reconstructed samples along different principal components in the feature space of learned classes. We follow the procedure done in (Dai et al., 2022b), calculating the principal components of the representations in each learned class, and then reconstructing the samples with representation closest to these principal components. Each row in Figure 4 displays objects of one class; each block of 5 images shows one principal component within each class. It clearly demonstrates that we may express the image diversity within each class by simply computing the principal components of the class. Even though our method does not use class label information, the model preserves statistical diversities between classes and within each class. We provide additional reconstructed images and feature space interpolation in Appendix D.

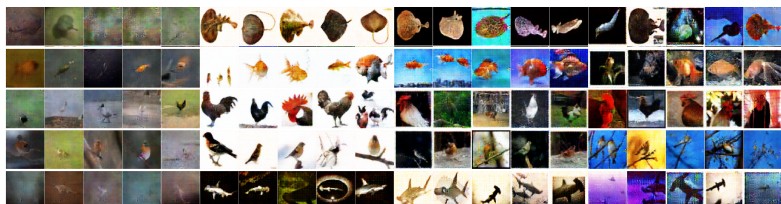

Figure 4: Five reconstructed samples $\hat{\boldsymbol{x}} = g(\boldsymbol{z})$ from $\boldsymbol{z}$'s closest to the subspace spanned by the top 4 principal components of learned features for 5 ImageNet classes (class "rajidae", "goldfish", "chicken", "bird", "shark").

## 4.3 GENERALIZABILITY TO AUTOENCODING UNSEEN DATASETS

To evaluate the generalizability of the learned model, we reconstruct samples of CIFAR-100 using a CSC-CTRL model which is only trained on CIFAR-10. Figure 5 shows a randomly reconstructed sample without cherry-picking. We observe that a lot of classes – for example, "lion", "wolf", and "snake" – which never appeared in CIFAR-10 can still be reconstructed, with high image quality. Moreover, if we visualize the samples along different principal components within the class, such as "bees" in Figure 6, we see that even the variance in the out-of-domain data samples may be captured by computing the principal components. It demonstrates that our model not only generalizes image reconstruction well to out-of-domain data, but also encodes a meaningful representation that preserves diversity between and within out-of-domain classes.

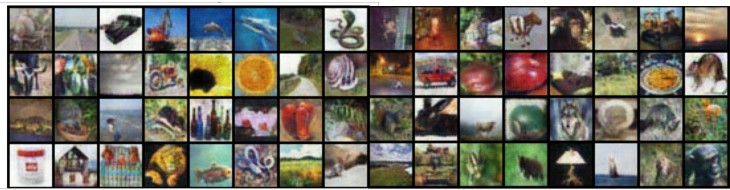

Figure 5: Visualization of randomly chosen reconstructed samples $\hat{\boldsymbol{X}}$ of CIFAR-100. The autoencoding model is only trained on the CIFAR-10 dataset.

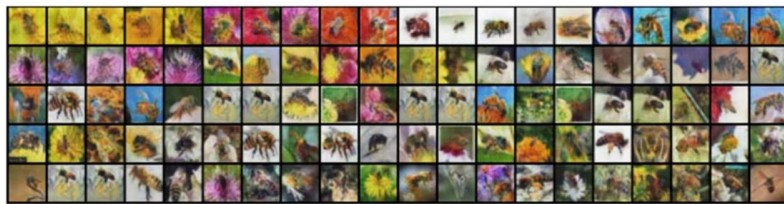

Figure 6: Five reconstructed samples $\hat{x} = g(z)$ from $z$'s closest to the subspace spanned by the top 20 principal components of learned features for CIFAR-100 class "bee". The model is only trained on CIFAR-10.

## 4.4 STABILITY OF CSC-CTRL

To test the stability of our method to input perturbation, we add Gaussian noise with mean of 0 to the original CIFAR-10 dataset. We use $\sigma$ to control the standard deviation of the Gaussian noise, i.e., the level of perturbation. Fundamentally, the difference between convolutional sparse coding layer and a simple convolutional layer is that the convolutional sparse coding layer assumes that the input features can be approximated by a superposition of a few atoms of a dictionary. This property of the convolutional sparse coding layer makes possible a stable recovery of the sparse signals with respect to input noise and, therefore, enables denoising (Elad, 2010; Wright & Ma, 2022). Hence, CSC-CTRL's autoencoding also functions as denoising of noisy data. We conducted experiments on CIFAR-10, with $\sigma = 0.5$, and STL-10, with a $\sigma = 0.1$. Because CIFAR-10 has a smaller resolution, we use a larger $\sigma$ so we can visualize the noise more clearly. From Figure 7, we see that CSC-CTRL outputs a better-denoised image. When noise level are larger, CSC-CTRL has a obvious advantage over CTRL, which uses traditional convolutional layers. We also present more quantitative analysis of denoising in Appendix E.1.

CIFAR-10                                  STL-10

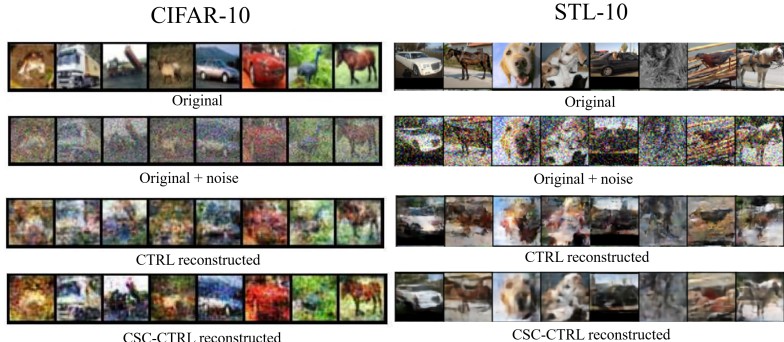

Figure 7: Denoising using CTRL and CSC-CTRL on CIFAR-10 with $\sigma = 0.5$ and STL-10 with $\sigma = 0.1$.

The CSC-CTRL model also demonstrates much better stability in training than the generic CTRL model. The coupling between the encoder and decoder makes the training more stable. For instance, the IS score of the CSC-CTRL model typically gradually increases and converges during training, whereas the CTRL model's IS score continuously drops after convergence. In addition, CSC-CTRL can converge with a wide range of batch sizes, from as small as 10 to as large as 2048, whereas CTRL can only converges with batch size larger than 512. These two properties are highly important from the perspective of engineering models within the CTRL framework. More details are in Appendix F.

## 5 CONCLUSION

In this work, we have shown with convincing evidence that within a closed-loop transcription framework, the classic and basic convolution sparse coding models are sufficient to construct good autoencoders for large sets of natural images. This leads to a simplifying and interpretable framework for learning and understanding the statistics of natural images. This new framework naturally integrates intermediate goals of seeking compact sparse representations with an end goal of obtaining an information-rich and structured representation, measured by the coding rate reduction. The learned models demonstrate generalizability and stability. We believe this gives a new powerful family of generative models that can better support a wide range of applications that require more interpretable and controllable image generation, reconstruction, and understanding.

## ETHICS STATEMENT

All authors agree and will adhere to the conference's Code of Ethics. We do not anticipate any potential ethics issues regarding the research conducted in this work.

## REPRODUCIBILITY STATEMENT

Settings and implementation details of network architectures, optimization methods, and some common hyper-parameters are described in the Appendix A.1. We will also make our source code available upon request by the reviewers or the area chairs.

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

# A   APPENDIX

## A.1   EXPERIMENT SETTINGS AND IMPLEMENTATION DETAILS

**Network backbones.** For CIFAR-10, we follow the 4-layers architecture which is used for MNIST in (Dai et al., 2022b), replacing all the standard convolutional layers with our drop-in convolutional sparse coding layers in Table 2 and 3 without extra modifications. Similarly, we adopt the 5-layers architecture for STL-10 (see Table 4 and 5) and ImageNet-1k (see Table 4 and 5).

| $z \in \mathbb{R}^{1 \times 1 \times 512}$ |
| --- |
| $4 \times 4$, stride=1, pad=0 CSC-inv BN 256 ReLU |
| $4 \times 4$, stride=2, pad=1 CSC-inv BN 128 ReLU |
| $4 \times 4$, stride=2, pad=1 CSC-inv BN 64 ReLU |
| $4 \times 4$, stride=2, pad=1 CSC-inv 1 Tanh |

Table 2: Decoder for CIFAR-10.

| RGB image $x \in \mathbb{R}^{32 \times 32 \times 3}$ |
| --- |
| $4 \times 4$, stride=2, pad=1 CSC 64 lReLU |
| $4 \times 4$, stride=2, pad=1 CSC BN 128 lReLU |
| $4 \times 4$, stride=2, pad=1 CSC BN 256 lReLU |
| $4 \times 4$, stride=1, pad=0 CSC 512 |

Table 3: Encoder for CIFAR-10.

| $z \in \mathbb{R}^{1 \times 1 \times 1024}$ |
| --- |
| $4 \times 4$, stride=1, pad=0 CSC-inv BN 512 ReLU |
| $4 \times 4$, stride=2, pad=0 CSC-inv BN 256 ReLU |
| $4 \times 4$, stride=2, pad=1 CSC-inv BN 128 ReLU |
| $4 \times 4$, stride=2, pad=1 CSC-inv BN 64 ReLU |
| $4 \times 4$, stride=2, pad=1 CSC-inv 1 Tanh |

Table 4: Decoder for STL-10 and ImageNet-1k.

| RGB image $x \in \mathbb{R}^{64 \times 64 \times 3}$ |
| --- |
| $4 \times 4$, stride=2, pad=1 CSC 64 lReLU |
| $4 \times 4$, stride=2, pad=1 CSC BN 128 lReLU |
| $4 \times 4$, stride=2, pad=1 CSC BN 256 lReLU |
| $4 \times 4$, stride=2, pad=0 CSC 512 |
| $4 \times 4$, stride=1, pad=0 CSC 1024 |

Table 5: Encoder for STL-10 and ImageNet-1k.

## A.2   OPTIMIZATION AND TRAINING DETAILS.

**General Settings.** Adam (Kingma & Ba, 2014) is adopted as the optimizer for all of our experiments. The hyper-parameters of Adam and the learning rate for each dataset will be discussed later in their own section. We choose $\epsilon^2 = 0.5$ for the maximin program (10) in all experiments, and the $\lambda$ inside the convolutional sparse coding layer is set to be 0.01 by default. For alternating minimizing and maximizing the objectives, we use the simple gradient descent-ascent algorithm. Most experiments are conducted on RTX 3090 GPUs.

**CIFAR-10.** For CIFAR-10, the learning rate is set to be $2 \times 10^{-4}$ with no decay, and we choose $\beta_1 = 0$, $\beta_2 = 0.9$ for Adam optimizer. Besides, we run 1000 epochs with mini-batch size 2000 for each experiment. In most cases, the model converges after about 300 epochs, with consistent visual quality and stable Inception Score.

**STL-10.** For STL-10, images are firstly resized to 64x64 using bilinear interpolation, and we run 1000 epochs with mini-batch size 1024, learning rate $2 \times 10^{-4}$ with no decay, and hyper-parameters $\beta_1 = 0.5$, $\beta_2 = 0.9$ for Adam optimizer. The model converges after about 300 epochs, with consistent visual quality and stable Inception Score.

**ImageNet-1k.** For ImageNet-1k, images are firstly center-cropped to 224x224 and then resized to 64x64 using bilinear interpolation during training. We run 10000 iterations with mini-batch size 128, learning rate $1 \times 10^{-4}$ and hyper-parameters $\beta_1 = 0.5$, $\beta_2 = 0.9$ for the Adam optimizer.

# B   ABLATION STUDY ON OPTIMIZATION STRATEGIES

In this section, we justify our choice of optimization strategy to optimize equation 10. We set the following optimization strategy as "Strategy 1", which was adopted in the original CTRL:

$$\max_{\theta(\boldsymbol{A})} \Delta R \text{ step}: \quad \boldsymbol{A}_{k+1} = \boldsymbol{A}_k + \lambda_{\max} \frac{\partial \Delta R}{\partial \theta} \cdot \frac{\partial \theta}{\partial \boldsymbol{A}} \Big|_{\boldsymbol{A}_k}, \tag{13}$$

$$\min_{\eta(\boldsymbol{A})} \Delta R \text{ step}: \quad \boldsymbol{A}_{k+2} = \boldsymbol{A}_{k+1} - \lambda_{\min} \frac{\partial \Delta R}{\partial \eta} \cdot \frac{\partial \eta}{\partial \boldsymbol{A}} \Big|_{\boldsymbol{A}_{k+1}}. \tag{14}$$

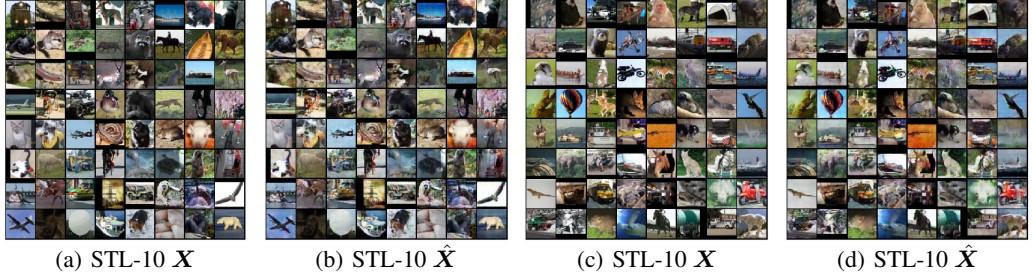

| (a) STL-10 $\boldsymbol{X}$ | (b) STL-10 $\hat{\boldsymbol{X}}$ | (c) STL-10 $\boldsymbol{X}$ | (d) STL-10 $\hat{\boldsymbol{X}}$ |
|---|---|---|---|

Figure 8: Visualizing the auto-encoding property of the learned CSC-CTRL ($\hat{\boldsymbol{X}} = g(f(\boldsymbol{X}, \theta), \eta)$) on STL-10. (Images are randomly chosen.)

We set the following optimization strategy as "Strategy 2", which is used in our work to optimize equation 10.

$$\max_{\theta(\boldsymbol{A})} \Delta R \ \text{step}: \quad \boldsymbol{A}_{k+1} = \boldsymbol{A}_k + \lambda_{\max} \frac{\partial \Delta R}{\partial \theta} \cdot \frac{\partial \theta}{\partial \boldsymbol{A}} \Big|_{\boldsymbol{A}_k}, \tag{15}$$

$$\min_{\boldsymbol{A}} \Delta R \ \text{step}: \quad \boldsymbol{A}_{k+2} = \boldsymbol{A}_{k+1} - \lambda_{\min} \Big( \frac{\partial \Delta R}{\partial \eta} \cdot \frac{\partial \eta}{\partial \boldsymbol{A}} + \frac{\partial \Delta R}{\partial \theta} \cdot \frac{\partial \theta}{\partial \boldsymbol{A}} \Big) \Big|_{\boldsymbol{A}_{k+1}}. \tag{16}$$

We run an ablation study on CIFAR-10 with hyper-parameters all the same from Appendix. A.1, except the training strategy. The results are shown in Table 6. Empirically, we found that Strategy 2 optimizes much better than Strategy 1.

Also in Table 6, we compare the performance of both strategies on the original CTRL model, where the encoder and decoder do not share weights. We observe that while Strategy 2 is clearly better than Strategy 1 for CSC-CTRL, it has less stellar performance on the original CTRL model, doing comparably well to Strategy 1. This indicates that Strategy 2 is at least as good as Strategy 1, roughly speaking, but only achieves significantly better performance in the case where the encoder and decoder share weights.

|  | CSC-CTRL | | CTRL | |
|---|---|---|---|---|
|  | IS ($\uparrow$) | FID ($\downarrow$) | IS ($\uparrow$) | FID ($\downarrow$) |
| Strategy 1 | 3.2 | 197.1 | 8.1 | 19.6 |
| Strategy 2 | 8.9 | 28.9 | 8.5 | 24.7 |

Table 6: Ablation study of CSC-CTRL on different optimization strategies through reconstructed image quality (IS/FID). $\uparrow$ means the higher the better. $\downarrow$ means the lower the better.

## C  MORE VISUALIZATION OF CSC-CTRL GENERATED IMAGES

Due to limited space in the main body, we show the generated images of STL-10 (see Figure 8) and some extra images of ImageNet (Figure 9). In this section. Fig 8 shows the auto-encoding properties of our learned framework on STL-10. Fig 9 shows a larger version of the reconstruction on ImageNet. We observe that even fine details in the image have been faithfully reconstructed, showcasing the power of our convolutional sparse coding network. Lastly, we include more generated images on ImageNet in Fig 10, demonstrating the image quality of our network.

## D  LINEAR INTERPOLATION IN THE LEARNED STRUCTURED FEATURE SPACE

Fig 11 shows reconstructed images whose features are linearly interpolated between pairs of images sampled from the ImageNet dataset. Formally, for two images $\boldsymbol{x}_1, \boldsymbol{x}_2$, the interpolated $\boldsymbol{x}$ is given by

$$\boldsymbol{x}_{\text{interp}} = g(\alpha f(\boldsymbol{x}_1) + (1 - \alpha) f(\boldsymbol{x}_2)) \tag{17}$$

where $\alpha \in [0, 1]$ varies in Fig 11 from 0 (on the left side) to 1 (on the right side).

The generated images show a continuous deformation from one sample to another. This verifies that our feature space is linearized and discriminative.

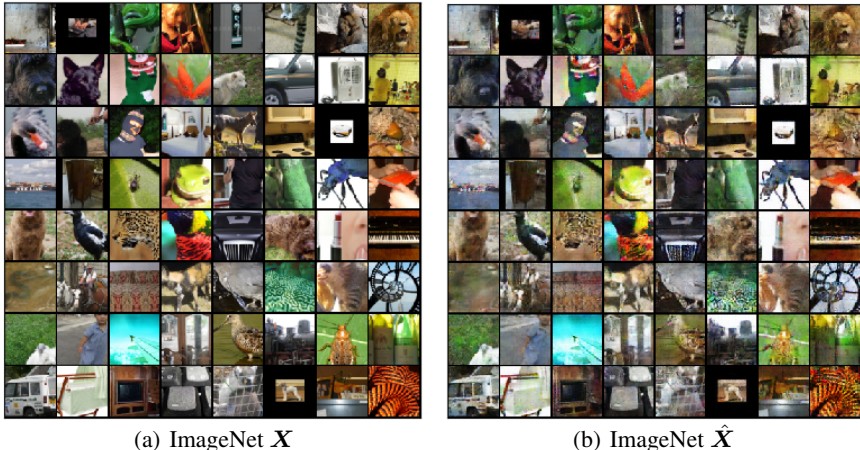

(a) ImageNet $X$                    (b) ImageNet $\hat{X}$

Figure 9: Visualizing the auto-encoding property of the learned CSC-CTRL ($\hat{X} = g(f(X, \theta), \eta)$) on ImageNet. (Images are randomly chosen.)

# E    MORE ANALYSIS OF DENOISING

## E.1    QUANTITATIVE MEASURE OF IMAGE DENOISING QUALITY

Due to space limitations in the main body, we present a quantitative analysis of denoising in this section. We use PSNR (Peak Signal-to-Noise Ratio), MSE (Mean Squared Error) and SSIM (Structural Similarity Index Measure) to measure the quality of denoising via CTRL and CSC-CTRL. Shown

| Noise level ($\sigma = 0.5$) | PSNR ($\uparrow$) | MSE ($\downarrow$) | SSIM ($\uparrow$) |
|---|---|---|---|
| CTRL | 13.3961 | 0.1914 | 0.1556 |
| CSC-CTRL | 17.0938 | 0.0837 | 0.3671 |

Table 7: Comparison of denoising via CTRL and CSC-CTRL with standard metrics. $\uparrow$ means the higher the better. $\downarrow$ means the lower the better.

in Table 7, CSC-CTRL performs significantly better than CTRL trained with the usual convolutional layers. It quantitatively verifies the effectiveness of the convolutional sparse coding layer for denoising.

## E.2    BETTER DENOISING THROUGH ADJUSTING $\lambda$

In fact, we can get better denoising effect by simply adjusting the $\lambda$ in the convolutional sparse coding layer in equation 6 **without** any additional training. Our default $\lambda$ is set to be 0.01 due to the scale between two objectives during the training stage. In the inference stage, we can further increase $\lambda$ to promote sparsity, which naturally leads to better denoising. From Table 8, we see that as $\lambda$ increases, CSC-CTRL generally improves at denoising.

## E.3    COMPARISON WITH OTHER DENOISING METHODS

To show how the CSC-CTRL model performs on denoising tasks, we compare our methods[7] with three denoising methods on the CIFAR-10 dataset with different noise levels in Table 9. We observe that our model achieves better performance than other methods which are specifically trained for denoising.

# F    STABILITY

In this section, we further verify the training stability of CSC-CTRL from two perspectives: mode collapse during training, and choice of batch size.

---

[7]The compared methods and their implementation can be found in `https://github.com/anushkayadav/Denoising_cifar10`

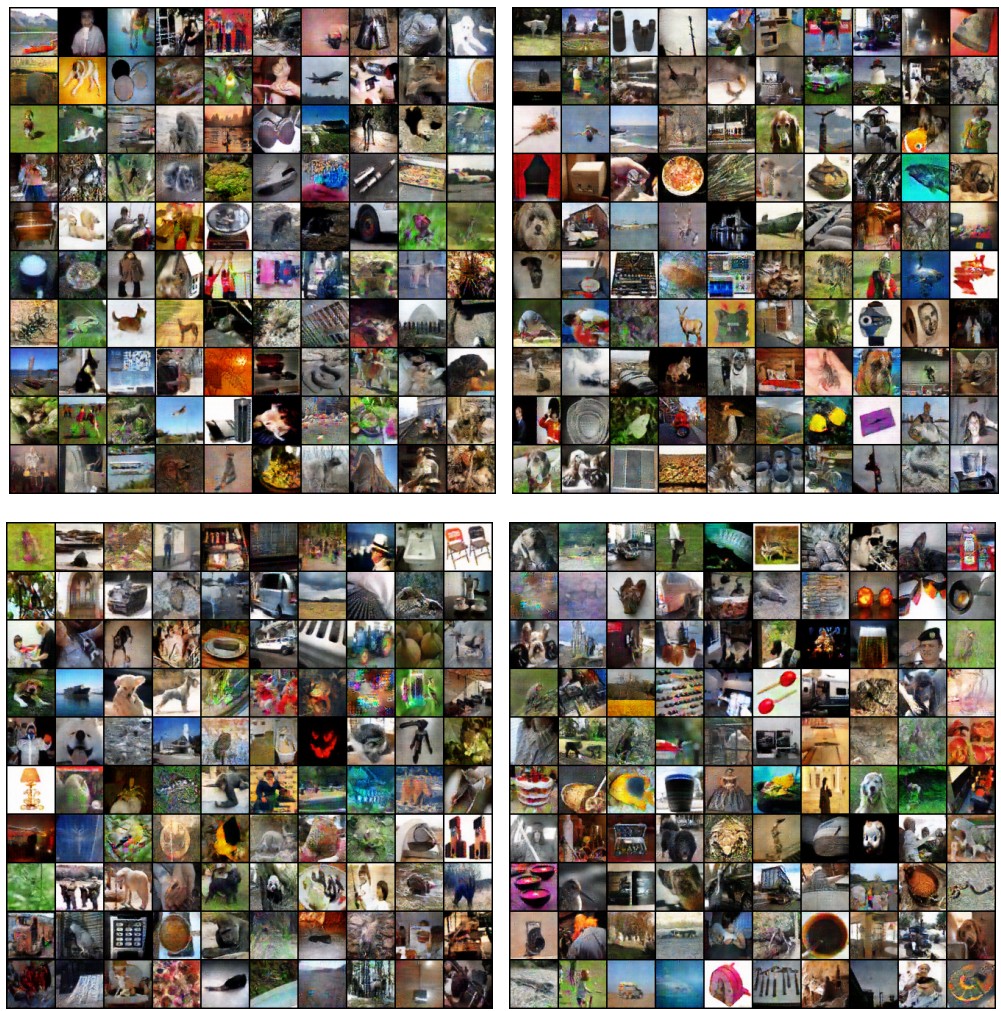

Figure 10: Visualizing randomly chosen reconstructed images of CSC-CTRL ($\hat{\boldsymbol{X}} = g(f(\boldsymbol{X}, \theta), \eta)$) on ImageNet.

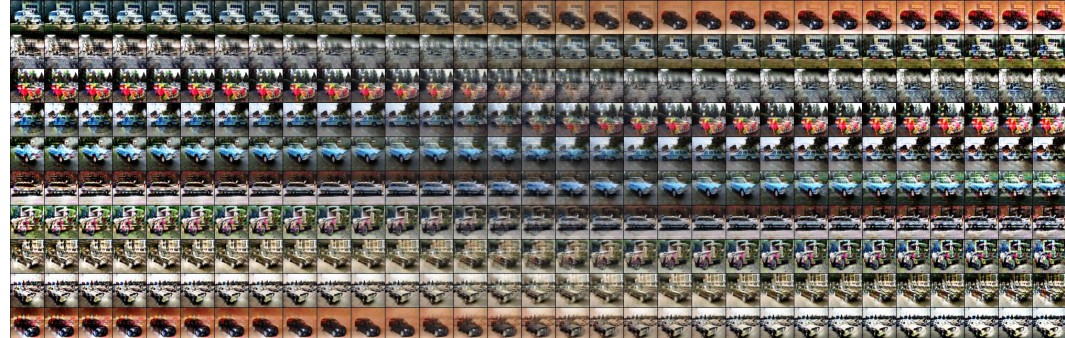

Figure 11: Images generated by features which were linearly interpolated in the learned feature space.

**Training Stability.** Experimentally, many previous methods such as CTRL and various GANs suffer from training instability. As shown in Figure 12, CTRL shows a clear training instability after 600 epochs. In contrast, CSC-CTRL training is much more stable, as the IS score barely drops. We conclude that CSC-CTRL suffers less from mode collapse.

| Noise level ($\sigma = 0.5$) | PSNR ($\uparrow$) | MSE ($\downarrow$) | SSIM ($\uparrow$) |
|---|---|---|---|
| $\lambda = 0.01$ (default) | 17.0938 | 0.0837 | 0.3671 |
| $\lambda = 0.1$ | 17.5774 | 0.0733 | 0.3955 |
| $\lambda = 0.2$ | 17.9926 | 0.0655 | 0.4222 |
| $\lambda = 0.3$ | 18.3500 | 0.0602 | 0.4479 |
| $\lambda = 0.4$ | 18.6068 | 0.0572 | 0.4658 |
| $\lambda = 0.5$ | **18.6155** | **0.0567** | **0.4676** |
| $\lambda = 0.6$ | 18.4205 | 0.0601 | 0.4593 |
| $\lambda = 0.7$ | 18.0563 | 0.0660 | 0.4364 |

Table 8: Comparison of denoising using different $\lambda$ with standard metrics. $\uparrow$ means the higher the better. $\downarrow$ means the lower the better.

| | PSNR ($\uparrow$) | MSE ($\downarrow$) | SSIM ($\uparrow$) |
|---|---|---|---|
| Noise level ($\sigma = 0.5$) | | | |
| CSC-CTRL | 17.094 | 0.084 | 0.367 |
| AE | 13.141 | 0.198 | 0.219 |
| AE w Sym | 15.784 | 0.117 | 0.318 |
| DnCNNs | 16.142 | 0.101 | 0.337 |
| Noise level ($\sigma = 0.1$) | | | |
| CSC-CTRL | 31.679 | - | 0.989 |
| AE | 24.830 | - | 0.868 |
| AE w Sym | 28.254 | - | 0.938 |
| DnCNNs | 28.992 | - | 0.947 |

Table 9: Comparison to three different denoising models on CIFAR10 with noise levels $\sigma = 0.1$ and $\sigma = 0.5$.

**Choice of Batch Size.** One notable flaw of the original CTRL (Dai et al., 2022b) is its reliance on a large batch size, normally greater than 512. This large batch size greatly increases the model's computational cost and limits its scalability. In Table 10, we compare whether each method converges under different batch sizes, from as small as 10 to as large as 2048. From the table, we see that CSC-CTRL can successfully converge on a wider range of batch sizes, even as low as 10. This greatly reduces the required computation power and enables easier training on more complicated datasets such as ImageNet.

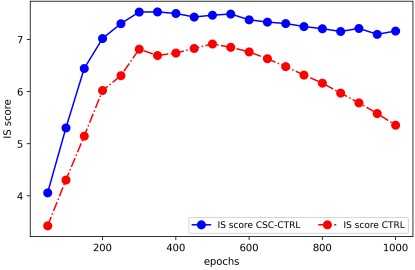

Figure 12: Training stability comparison of CTRL and CSC-CTRL with IS score on CIFAR-10.

# G  SENSITIVITY TO CHOICE OF RANDOM SEED

We report in Table 11 the IS/FID of CSC-CTRL with different random seeds. The experiments are conducted on CIFAR-10. As we can see, the choice of random seed has very little effect on the performance of CSC-CTRL.

| Batch Size | 10 | 64 | 128 | 256 | 512 | 1024 | 1600 |
|---|---|---|---|---|---|---|---|
| CSC-CTRL | ✓ | ✓ | ✓ | ✓ | ✓ | ✓ | ✓ |
| CTRL | ✗ | ✗ | ✗ | ✗ | ✓ | ✓ | ✓ |

Table 10: Comparison of CTRL and CSC-CTRL trained with different batch sizes. ✓ means the method has successfully converged, ✗ means the method fails to converge.

| Random Seed | 1 | 5 | 10 | 15 | 100 |
|---|---|---|---|---|---|
| IS | 8.8 | 8.7 | 8.9 | 8.8 | 8.9 |
| FID | 28.9 | 27.9 | 28.5 | 28.1 | 28.6 |

Table 11: Ablation study on varying random seeds.

## H  VISUALIZING THE LEARNED DICTIONARIES

In Figure 13, we provide a visualization of the learned dictionaries of all layers of CSC-CTRL trained on ImageNet. The kernel size for all layers is $4 \times 4$. The dimension of the dictionary in the first layer of CSC-CTRL is $64 \times 3 \times 4 \times 4$. We visualize this dictionary as $64 \cdot 3 = 192$ patches of size $4 \times 4$, arranged into a $14 \times 14$ grid and displayed in grayscale. For the layers after the first layer, there are more than $14 \cdot 14 = 196$ kernels; we visualize the first 196 kernels for readability.

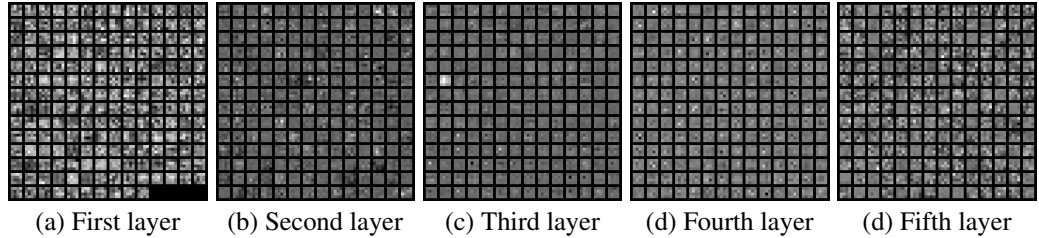

(a) First layer    (b) Second layer    (c) Third layer    (d) Fourth layer    (d) Fifth layer

Figure 13: Visualization of the learned dictionary of each layer of CSC-CTRL trained on ImageNet.

## I  ABLATION STUDY ON CSC LAYER BASED AUTOENCODERS

In this section we demonstrate how the CSC layer may benefit autoencoding, especially with respect to generalizing to unseen datasets. We use the architecture from Table 2 and Table 3 to get a CSC-layer based autoencoder, which we train in several ways. First, we directly use mean square error loss on the input $x$ of the encoder and output $\hat{x} = g(f(x))$ of the decoder to train the model; we call this "CSC-AE". Then, we add the $MCR^2$ loss to "CSC-AE" to get the "CSC-sAE" method. Figure 14 shows the reconstructed performance of these methods and their generalizability to CIFAR-100.

On the other hand, to fairly evaluate the influence of the CSC layer to reconstruction performance, we replace it with a convolutional layer. More precisely, we use the architecture from Table 2 and Table 3 but replace the CSC layers of the encoder with convolutional layers, and also replace all CSC-inv layers of the decoder with ConvTranspose2D layers to get a CNN-based autoencoder. With the generic mean square error loss, we get the "Conv-AE" model; with VAE loss, we get the "Conv-VAE" model. Finally, "Conv-CTRL" is the same as the original CTRL method. Figure 15 shows the reconstruction performance of these methods and their generalizability to CIFAR-100.

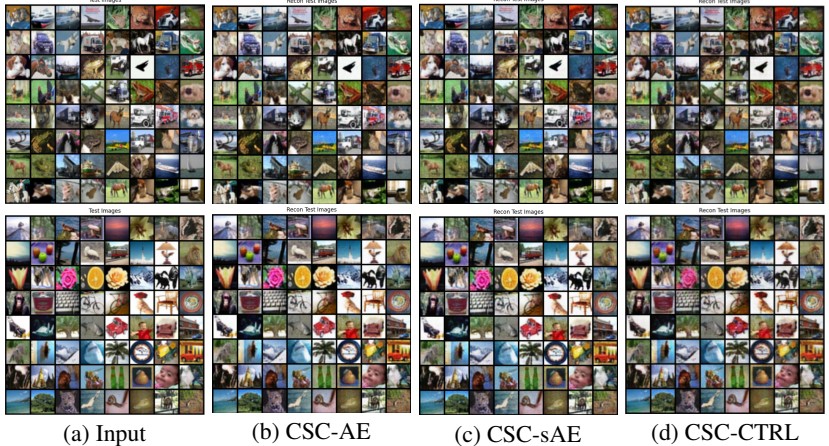

(a) Input    (b) CSC-AE    (c) CSC-sAE    (d) CSC-CTRL

Figure 14: Image reconstruction results of CSC layer based autoencoder trained on CIFAR-10 with different loss function such (b) MSE, (c) MSE+MCR$^2$, and (d) closed loop framework. The first raw shows the reconstructed results on CIFAR-10 test set. The second raw shows the reconstructed results on CIFAR-100 test set, which also reflects the generalizability.

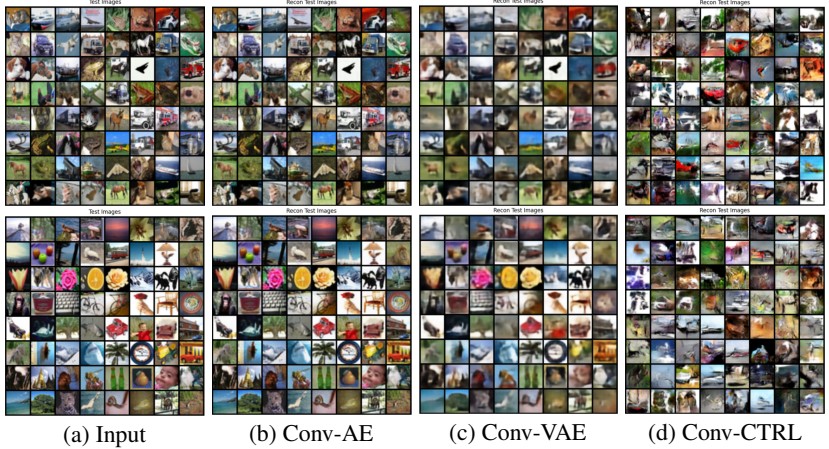

(a) Input    (b) Conv-AE    (c) Conv-VAE    (d) Conv-CTRL

Figure 15: Image reconstruction results of Convolution layer based autoencoder trained on CIFAR-10 with different loss function such (b) MSE, (c) MSE+KL, and (d) closed loop framework. The first raw shows the reconstructed results on CIFAR-10 test set. The second raw shows the reconstructed results on CIFAR-100 test set, which also reflects the generalizability.

| Method | Model Size | Train Time | CIFAR-10 | | STL-10 | | ImageNet | |
|---|---|---|---|---|---|---|---|---|
| | | | IS↑ | FID↓ | IS↑ | FID↓ | IS↑ | FID↓ |
| *GAN based methods* | | | | | | | | |
| DCGAN (Radford et al., 2015) | | | 6.6 | 35.3 | 7.8 | - | - | - |
| SNGAN (Miyato et al., 2018) | | | 7.4 | 29.3 | 9.1 | 40.1 | 7.3 | 48.7 |
| *VAE based methods* | | | | | | | | |
| VAE (Kingma & Welling, 2013) | | | 5.2 | 55.9 | - | - | - | - |
| NVAE (Vahdat & Kautz, 2020) | 10M | >55 h | - | 50.8 | - | - | - | - |
| NVAE (Recon) | 10M | >55 h | - | 2.67 | - | - | - | - |
| DCVAE (Parmar et al., 2021) | 4M | >24h | 8.2 | 17.9 | 8.1 | 41.9 | - | - |
| DCVAE (Recon) | 4M | >24h | 7.9 | 21.4 | 8.4 | 43.6 | - | - |
| *Flow based methods* | | | | | | | | |
| GLOW (Kingma & Dhariwal, 2018) | | | - | 46.9 | - | - | - | - |
| Residual Flow (Chen et al., 2019) | | | - | 50.8 | - | - | - | - |
| *CTRL based methods* | | | | | | | | |
| CTRL (Dai et al., 2022b) | 1.0M | 15 h | 8.1 | 19.6 | 8.4 | 38.6 | 7.7 | 46.9 |
| CSC-CTRL (ours) | 0.5M | 8 h | 8.9 | 28.9 | 9.1 | 48.1 | 12.5 | 34.5 |

Table 12: Comparison on CIFAR-10, STL-10, and ImageNet-1K. The network architectures used in CSC-CTRL are 4-layers for CIFAR-10, 5-layers for STL-10 and ImageNet respectively which are much smaller than other compared methods. NVAE(recon) means the results of reconstruction, the column "Train Time" means the hours the model used for training.

