# OpenReview forum: "Closed-loop Transcription via Convolutional Sparse Coding"
_ICLR.cc/2023/Conference — Submitted to ICLR 2023_

### Official Review · Reviewer_qbVZ · 2022-10-24

**Confidence:** 4
**Clarity, Quality, Novelty And Reproducibility:** See my comments above.
**Correctness:** 3
**Technical Novelty And Significance:** 3
**Empirical Novelty And Significance:** 2
**Recommendation:** 6

**Strength And Weaknesses:**

This paper is interesting and represents a valuable contribution. Indeed, it is nice to see models based on CSC providing such nice results. The paper is also clear and easy to read.

This reviewer does however have a few comments and concerns:

Main comments:

1. This main paper has very few details about the implementation or on how these networks are implemented, and all of these are deferred to the appendix. While this is mostly fine, it would be nice to have some clarifications. Chiefly: from the description of the decoder, it seems like every decoding process of each layer is completely linear. Is this correct? If this is the case, then is the entire decoder (for all layers) completely linear? If this is the case, can the authors explain why not to replace the product of all these linear layers with simply a (single) linear layer, as they can indeed be made equivalent?

2. An important question refers to the way the authors optimize the problem in Eq. (9). The authors parameterize two players (encoder and decoder), and pose this as a max-min problem. Both networks are parameterized by the dictionary parameters, $A$. Now, instead of optimizing this problem in this way, in the line immediately following Eq.(12), the authors comment that, empirically, they find it beneficial to modify one of the first order updates in some pretty ad-hoc way. They also comment in passing that they include an ablation study on this choice in Appendix B. Appendix B indeed shows the comparison of their performance with their 'original' motivated method (strategy 1) versus the 'ad-hoc update' version (strategy 2). My issue with this is that Strategy 2 does not just improve a little their method giving them a small boost -which would be reasonable- but instead fully determines the competitive advantage of their method: Their strategy 2 performs almost an order of magnitude better (3.2 vs 8.9 in IS, and 197 vs 28.9 in FID) for CIFAR-10. Moreover, the "worst" performing method that they compare with (VAE) achieve 5.2 (IS) and 56 (FID), which is about 3 times (say, on average) as good as their strategy 1.
In an empirical paper like this, it is natural to have some small details of the implementation that can provide a small improvement. Yet, from these numbers, it appears as this little ad-hoc trick is responsible for their method becoming "state-of-the-art", and I think that as such it should deserve some closer study.

3. It is also very strange that their method provides sample-wise alignment: indeed, as the authors mention, the entire network is trained on a loss that looks at statistics of the distribution of (real and generated) samples. Thus, it indeed need to provide matched samples, as shown by [Dai et al, 2022]. In this work, the authors seem to use the same loss and training framework, but all of a sudden they can reconstruct matched samples. How come?

4. While the contribution of this paper is interesting and relevant, in my humble opinion and respectfully, the paper is a bit on the verbose side and some comments are a bit too subjective or exaggerated. For instance, in a couple of places, the authors comment that their methods shows "striking performance on large datasets", or "splendid visual quality". It is true that their IS and FID numbers are better than SNGAN and CTRL in ImageNet, but looking at the generated images (e.g. in Fig 3d and Fig 9b), I would not qualify their quality as "striking" or "splendid" (plenty of artifacts, both shape-wise and color-wise).

5. To the best of my knowledge, the first work in relating sparse coding networks to deep networks, and even suggesting replacing each convolutional layer by a lasso formulation -as done in this work- is the work by Papyan et al, 2017, in JMLR. I think credit is due to that work for some of the ideas appearing here.

I enjoyed reading this paper, and I look forward to reading the authors' responses to clarify my understanding.


Small comments:

- Why do the authors refer to (de)convolution layers? Looking at their definitions, what they refer to "(de)convolution" is just a cross-correlation operator - which is just the adjoint of a convolution operator.

- I'm curious why the approach chose to abbreviate "closed-loop transcription" as "CTRL". Did the authors perhaps meant "CLTR"?

**Summary Of The Paper:**

This paper proposes an interesting approach to design autoencoders based on convolutional sparse coding. In this framework, each layer-wise operation is replaced by an approximate solution to a Lasso problem, with a convolutional dictionary. The reconstruction ('decoding') from the so-obtained sparse codes is given simply by a linear model. The authors train the resulting architecture with a recently proposed method, or loss function, termed Closed Loop Transcription (CTRL), which optimizes the rate reduction of encoder and decoder in a minimax formulation. The paper presents empirical results on CIFAR 10, CIFAR 100 and Imagenet 1K.


**Summary Of The Review:**

Interesting paper with potentially valuable contribution, and with some elements that need further clarification.

---

> ### Author Response · Authors · 2022-11-19
> **Response to Q1~Q2 of Reviwer qbVZ**
>
> We thank the reviewer for their insightful comments.
>
>
> > Q1. This main paper has very few details about the implementation or on how these networks are implemented, and all of these are deferred to the appendix. While this is mostly fine, it would be nice to have some clarifications. Chiefly: from the description of the decoder, it seems like every decoding process of each layer is completely linear. Is this correct? If this is the case, then is the entire decoder (for all layers) completely linear? If this is the case, can the authors explain why not to replace the product of all these linear layers with simply a (single) linear layer, as they can indeed be made equivalent?
>
> A: Thank you for pointing it out. We have added this detail to the main body, but it is also in Table 2 to Table 5 in Appendix. We still have non-linear operators in the decoder (ReLU, BatchNorm). Hence, it is not completely linear.
>
> > Q2. An important question refers to the way the authors optimize the problem in Eq. (9). The authors parameterize two players (encoder and decoder), and pose this as a max-min problem. Both networks are parameterized by the dictionary parameters, A. Now, instead of optimizing this problem in this way, in the line immediately following Eq.(12), the authors comment that, empirically, they find it beneficial to modify one of the first order updates in some pretty ad-hoc way. They also comment in passing that they include an ablation study on this choice in Appendix B. Appendix B indeed shows the comparison of their performance with their 'original' motivated method (strategy 1) versus the 'ad-hoc update' version (strategy 2). My issue with this is that Strategy 2 does not just improve a little their method giving them a small boost -which would be reasonable- but instead fully determines the competitive advantage of their method: Their strategy 2 performs almost an order of magnitude better (3.2 vs 8.9 in IS, and 197 vs 28.9 in FID) for CIFAR-10. Moreover, the "worst" performing method that they compare with (VAE) achieve 5.2 (IS) and 56 (FID), which is about 3 times (say, on average) as good as their strategy 1. In an empirical paper like this, it is natural to have some small details of the implementation that can provide a small improvement. Yet, from these numbers, it appears as this little ad-hoc trick is responsible for their method becoming "state-of-the-art", and I think that as such it should deserve some closer study.
>
> A: Thank you for the reviewer’s comments. First, we note that the “max-min game” where both players optimize over the same variable (i.e. $A$) is not a well-defined game, and so the pre-existing theory of CTRL from [Pai et al, 2022] breaks down. Instead, we try an approach which preserves most of the qualitative essence of CTRL: alternating updates which maximize the dictionary’s expressive power (i.e. keeping the dictionary non-degenerate and fitting the structure of the data) and ensure that the whole network is invertible. Both optimization strategies fall under this conceptual framework. The difference is that the second strategy uses both the encoder and decoder information in order to tune the network weights to make the network invertible, while the first strategy only uses the decoder information.
> Suppose that at a given iteration, say timestep $t$, where the encoder $f_{t}$ is not perfectly invertible by the decoder $g_{t}$, we just use the decoder gradient information to update weights. Then, given small enough step size, the weights $A_{t}$ will be updated (to, say, $A_{t + 1}$) so that the decoding $g_{t + 1}$ works better, very crucially on the encoder $f_{t}$ which is induced by the weights $A_{t}$. But since the encoder and decoder share weights, such an update will also affect the encoder, and so it’s not even conceptually true that $g_{t + 1}$ will invert $f_{t + 1}$ better than $g_{t}$ can already invert $f_{t}$. This issue is ameliorated if we use both encoder and decoder gradient information to update weights. This perspective is confirmed by our ablation study in Table 6 of the Appendix B, which demonstrates that the strategy we use is only significantly better when the weights are shared (as opposed to when the weights are not shared, i.e., in  regular CTRL).
> Note that the above argument glosses over a lot of complexity about the non-convex optimization, and is only meant to be an intuition.

---

> ### Author Response · Authors · 2022-11-19
> **Response to Q3~Q7 of Reviewer qbVZ**
>
> > Q3. It is also very strange that their method provides sample-wise alignment: indeed, as the authors mention, the entire network is trained on a loss that looks at statistics of the distribution of (real and generated) samples. Thus, it indeed need to provide matched samples, as shown by [Dai et al, 2022]. In this work, the authors seem to use the same loss and training framework, but all of a sudden they can reconstruct matched samples. How come?
>
> A: Thank you for pointing it out. Please refer to eq (6) of the main body of the paper. Each layer in the encoder solves the given LASSO problem, which enforces sample-wise consistency by penalizing the L2 norm between the layer inputs and their reconstructions from the dictionary and the output sparse codes. Hence, since each layer enforces sample-wise consistency, the whole network enforces sample-wise consistency, and the overall reconstruction results are much better than that of the original work [Dai et al, 2022].
>
> > Q4. While the contribution of this paper is interesting and relevant, in my humble opinion and respectfully, the paper is a bit on the verbose side and some comments are a bit too subjective or exaggerated. For instance, in a couple of places, the authors comment that their methods show "striking performance on large datasets", or "splendid visual quality". It is true that their IS and FID numbers are better than SNGAN and CTRL in ImageNet, but looking at the generated images (e.g. in Fig 3d and Fig 9b), I would not qualify their quality as "striking" or "splendid" (plenty of artifacts, both shape-wise and color-wise).
>
> A: Thank you for the comments. Here, our overall message is that, compared to other sparse coding-based generative models, our method has “striking” results; in particular, while our method is comparable to the state-of-the-art, other sparse coding-based models cannot even scale up to large datasets such as ImageNet. We changed the words and made the claim clear in the revised version.
>
> > Q5. To the best of my knowledge, the first work in relating sparse coding networks to deep networks, and even suggesting replacing each convolutional layer by a lasso formulation -as done in this work- is the work by Papyan et al, 2017, in JMLR. I think credit is due to that work for some of the ideas appearing here.
>
> A: Thank you for the comments. We cited it in the revised version.
>
>
> > Q6. Why do the authors refer to (de)convolution layers? Looking at their definitions, what they refer to "(de)convolution" is just a cross-correlation operator - which is just the adjoint of a convolution operator.
>
> A: Thank you for pointing it out. Please refer to the reference [Zeiler et al, 2010; Zeiler et al, 2011] in the revised version.
>
> > Q7. I'm curious why the approach chose to abbreviate "closed-loop transcription" as "CTRL". Did the authors perhaps meant "CLTR"?
>
> A: Thank you for pointing it out. We just follow the original CTRL paper’s name. From the paper, the name is “Closed loop TRanscription to Ldr (CTRL)”.

---

> ### Author Response · Authors · 2022-11-25
> **Further Discussion with the Reviewer**
>
> Dear Reviewer qbVZ
>
> We thank you for the precious review time and valuable comments. We have provided corresponding responses and results, which we believe have covered your concerns. We hope to further discuss with you whether or not your concerns have been addressed. Please let us know if you still have any unclear parts of our work.

---

> > ### Comment · Reviewer_qbVZ · 2022-11-28
> > **Thank you for the clarifying responses**
> >
> > Dear authors,
> >
> > Thank you for your clarifying responses (and excuses for the latency, as I tended to some personal matters).
> >
> > The responses do clarify and answer my questions. I am now slightly increasing my recommendation from 5 to 6. The reason I cannot be more supportive of this work is because it seems that the optimization aspects of the learning problem (in Eq 9) are not quite clear and based only on coarse intuition. Since, as the authors say, the mix-max problem in Eq (9) "is not a well defined game", I wonder what loss the authors do minimize in the end. In my humble opinion, this would be fine if these ad-hoc tricks gave a slight increase in performance, but here it seems to really determine the overall performance of the method.
> >
> > A small follow-up on the comparison with other "sparse coding-based generative models": the current proposed method, as the authors now clarify, not only has sparse coding layers but also ReLU networks and batch-norm layers in between each layer. Thus, this is more complex than previous "sparse coding based generative models", resulting in an amalgam of different non-linear functions that should not be dubbed only as "sparse coding networks". It'd be great if the authors could stress this, as it seems that these non-linear functions (that are unrelated to sparse coding) seem to be crucial for good performance.
> >
> > In summary: I believe this is an interesting contribution. I think a clearer understanding of the learning formulation would really improve this paper further.

---

> > > ### Author Response · Authors · 2022-12-08
> > > **Further discussion with Reviewer qbVZ**
> > >
> > > We thank the reviewer for the valuable comment! We will conduct a theoretical analysis on the optimization in future work. We'll also stress the non-linear portion of the network in future edition since we cannot change the text now. It has been a great pleasure having you review our work!

---

### Official Review · Reviewer_Ti8A · 2022-10-25

**Confidence:** 4
**Correctness:** 3
**Technical Novelty And Significance:** 1
**Empirical Novelty And Significance:** 2
**Recommendation:** 6

**Clarity, Quality, Novelty And Reproducibility:**

The paper is written clearly. However, there are certain statements that need clarification. See my comments above. Quality is ok. However, the paper is not original. See my comments on novelty above.

**Strength And Weaknesses:**

strengths:

The paper was easy to read. It is organized, and I could follow the arguments. The experiments seem reproducible (enough details are provided in the appendix). There are reasonable visualizations and characterizations of their method. However, given my concerns explained below, I do not recommend an acceptance.

weaknesses:

- Although this paper combines two interesting ideas, it lacks novelty. To elaborate, the paper does not offer new knowledge. The ideas are: constructing autoencoders based on an optimization model (in this case the deep generative sparse coding model), and using CTRL to train the network to boost up image generation quality of the network.

- The literature review is not thorough. For example, unrolling optimization algorithms are used abundantly in the literature. [1] is a review paper about this approach and its wide usage in image and signal processing applications. For networks-based convolutional sparse coding, [2,3], among the many, is missing; the works show how to construct a recurrent autoencoder based on a shallow sparse coding model and train it for image denoising. Both of these papers have already shown the computational and trainable parameter efficiency of unrolling approach and have shown the competitive performance against SOTA at the time of their publications.

- Compared to other image generation methods, it is not clear what this paper is offering. A thorough discussion and comparison with SOTA are missing. Or a discussion on why SOTA comparison is not needed. This is especially needed as the proposed method is a combination of two already existing approaches. Moreover, this is particularly important, as the authors argue that they are the first to scale up CSC.

- Lack of baselines: Although I enjoyed looking into the reconstructed images, baselines are lacking for several visualizations. The principal approach for image generation can also be applied to other generating architecture. How is their performance (e.g., Figure 4,6)? For example, in Section 4.3, the authors argue that their framework generalizes to autoencoding unseen datasets. Aren't other autoencoders able to do this? If yes/no, please include such discussion and comparison (e.g., for Figure 5).

- Interpretability: the authors emphasize that their framework has the benefit of interpretable representation. However, they do not define interpretability. Indeed, there are many definitions of interpretability [4], and elaboration is needed here. Regarding the interpretability of deep networks constructed based on the generic sparse coding model, [5] is missing in their literature review. Certain questions arise about the interpretability, the kernel size in the trained architecture is too small. Hence, in terms of learned features, their kernel seems rather small to learn human interpretable features. One question is how the interpretability of their framework is useful.

- The paper confuses two concepts of "image generation" and "image reconstruction". In many of the examples, for example, Figure 2, the authors are visualizing the network reconstruction (pass the image into the encoders, and then decoder). This does not image generation; the code is created by passing an image into the encoder.

Some questions and recommendations.

- In Section 4.2, the authors say that "... reconstructing the samples with representation closest to these principal components". Where are the samples coming from? From my understanding, there seems to be a set of encodings from a set of images. The method is using those encodings that are closest to the principal components of the representations to reconstruct an image. Again, there is no sampling as in GANs. This is not an image generation, and looking into PC of a set of representations can be applied to other autoencoders. Please elaborate.

- What is the significance of large datasets that the author emphasizes if the goal is not classification, but image reconstruction? At the end of the day, all networks are being trained by gradient descent (hence not all data is going to be used at once for parameter updates). I do find the contribution of the fact that they are the first to train on imagenet-1k for image reconstruction (not classification) to be a minor contribution.

- I recommend adding SOTA to Table 1, to give the reader a perspective on how far the performance is from SOTA. Perhaps to help you with your arguments on scaling up csc but not comparing to SOTA, I suggest including an extra column to report on network parameters and computational efficiency.

- The paper frequently makes statements that are not precise and need elaboration. For example, prior to Section 3, it states "... have several good benefits unknown to any of the previous generative methods". Please be precise and explain the benefit and specify prior works.

- Please report the number of parameters to support the claim on efficiency.

Minor

1. Page 2, bullet 1, citation is missing for previous sparse coding works.
2. The authors emphasize much on convolutional and miss the generic sparse coding literature (at the end of the day, convolution is a linear operator with structure). In the unrolling literature, there are many theoretical works on unrolled sparse coding that may worth to be cited. Here are two [6,7].
3. replace "." with "," on (4).
4. It would be nice to provide an additional explanation on CTRL. For example, their method from the perspective of fixed points.
5. Suggest changing \lambda_max and \lambda_min in (11) and (12) to avoid confusion with \lambda in the lasso.
6. please cite lasso [8] before (6).
7. Please report the noise \sigma of the image in the scale of a/255. What is a? (Section 4.4)
8. There are certain wordings not supported in the paper. Please remove or support. E.g., in conclusion, the usage of "convincing" and "unprecedented".
9. Figure 8, if (a) and (c) are similar, please combine. same for (b) and (d).
10. I liked the stability discussion from the perspective of the connectivity of the encoder and decoder. If space allows, I recommend moving the losses from the appendix into the main paper.

[1] Monga V, Li Y, Eldar YC. Algorithm unrolling: Interpretable, efficient deep learning for signal and image processing. IEEE Signal Processing Magazine. 2021.
[2] Simon D, Elad M. Rethinking the CSC model for natural images. Advances in Neural Information Processing Systems. 2019.
[3] Tolooshams B, Song A, Temereanca S, Ba D. Convolutional dictionary learning based auto-encoders for natural exponential-family distributions. In International Conference on Machine Learning 2020.
[4] Doshi-Velez F, Kim B. Towards a rigorous science of interpretable machine learning. arXiv preprint arXiv:1702.08608. 2017.
[5] Tolooshams B, Ba D. Stable and interpretable unrolled dictionary learning. Transaction in Machine Learning Research. 2022.
[6] Chen X, Liu J, Wang Z, Yin W. Theoretical linear convergence of unfolded ISTA and its practical weights and thresholds. Advances in Neural Information Processing Systems. 2018.
[7] Ablin P, Moreau T, Massias M, Gramfort A. Learning step sizes for unfolded sparse coding. Advances in Neural Information Processing Systems. 2019.
[8] Tibshirani R. Regression shrinkage and selection via the lasso. Journal of the Royal Statistical Society: Series B (Methodological). 1996.

**Summary Of The Paper:**

The paper proposes an autoencoder architecture whose encoder and decoder are constructed based on the convolutional sparse coding generative model. The encoder is a convolutional sparse coding layer solving lasso (sparse coding problem) by unrolling FISTA (fast iterative thresholding algorithm). The decoder is dictated by the sparse coding/dictionary learning model. In place of the usual reconstruction loss, the authors use closed-loop transcription (CTRL) to train the network, i.e., to maximize the rate reduction of the learned sparse codes. To learn the dictionary, they use a max-min optimization problem for the encoder to discriminate between image encoding and reconstruction encoding and for the decoder (generator) to minimize the difference.

Here are their contributions from their perspective: The paper argues that they are the first to scale up the sparse coding model and train it with ImageNet. They argue that one of the benefits of their network architecture is having interpretable and structured representation. This arises due to their sparse coding encoder architecture and their training approach of maximizing the information gain. With the usage of the CTRL approach to train the framework, they offer sample-wise alignment. They emphasize that the construction of a network-based of sparse coding model offers efficient model and training. They focus on the image generation quality of the framework and how it generalizes.

Here is their contribution from the reviewer's perspective: The paper combines two existing ideas of unrolled networks and CTRL training. They show the reconstruction/generation performance of the method.

**Summary Of The Review:**

The paper lacks a clear novelty. It combines two ideas and the majority of their findings are already known in each of those ideas (unrolling optimization method to create autoencoder, and CTRL to train for image generating autoencoder). Aside from novelty, the method although argued to be scaled up of CSC, still does not choose SOTA as a baseline. Literature review missing key papers. See my review above.


--------------

My original score was 5. I increased it to 6 after a long discussion.

---

> ### Author Response · Authors · 2022-11-19
> **Response to Q1~Q5 of Reviewer Ti8A**
>
> We thank the reviewer for their insightful comments.
>
> > Q1. Although this paper combines two interesting ideas, it lacks novelty. To elaborate, the paper does not offer new knowledge. The ideas are: constructing autoencoders based on an optimization model (in this case the deep generative sparse coding model), and using CTRL to train the network to boost up image generation quality of the network.
>
> A: This is a common question; other reviewers have asked it, and so we will respond in a separate (global) response at the top of the review page.
>
> > Q2. The literature review is not thorough. For example, unrolling optimization algorithms are used abundantly in the literature. [1] is a review paper about this approach and its wide usage in image and signal processing applications. For networks-based convolutional sparse coding, [2,3], among the many, is missing; the works show how to construct a recurrent autoencoder based on a shallow sparse coding model and train it for image denoising. Both of these papers have already shown the computational and trainable parameter efficiency of unrolling approach and have shown the competitive performance against SOTA at the time of their publications.
>
> A: [2] are targeting the denoising task for applications; they do not particularly target our image reconstruction task, and they do not attempt to scale up to ImageNet. There are also have some follow up works to handle the image reconstruction task, which we cited and discussed in the related works section. [1] is a good review paper, while [3] is a recent paper about this topic; we added both citations to the revision.
>
> > Q3. Compared to other image generation methods, it is not clear what this paper is offering. A thorough discussion and comparison with SOTA are missing. Or a discussion on why SOTA comparison is not needed. This is especially needed as the proposed method is a combination of two already existing approaches. Moreover, this is particularly important, as the authors argue that they are the first to scale up CSC.
>
> A: Thank you for pointing it out. The main contribution of this work is to make the sparse coding-based autoencoder work on large scale datasets like ImageNet. We know that, although our method is close to SOTA and has good performance, it’s not yet at SOTA level. Possible reasons could be that we use smaller network size and don’t use any engineering tricks, while SOTA methods use larger networks and are well-engineered on large datasets. We added the model size and training time of other compared methods in the revised version (see Table 12). This will give the reader an idea of how efficient our model is compared to SOTA methods.
> Also, in the first paragraph of Section 4, we elaborated on why SOTA comparison is unnecessary.
>
>
> > Q4. Lack of baselines: Although I enjoyed looking into the reconstructed images, baselines are lacking for several visualizations. The principal approach for image generation can also be applied to other generating architectures. How is their performance (e.g., Figure 4,6)? For example, in Section 4.3, the authors argue that their framework generalizes to autoencoding unseen datasets. Aren't other autoencoders able to do this? If yes/no, please include such discussion and comparison (e.g., for Figure 5).
>
> A: Thank you for pointing it out. We add discussion regarding this point to Section I of the Appendix in the revised version.
>
> > Q5. Interpretability: the authors emphasize that their framework has the benefit of interpretable representation. However, they do not define interpretability. Indeed, there are many definitions of interpretability [4], and elaboration is needed here. Regarding the interpretability of deep networks constructed based on the generic sparse coding model, [5] is missing in their literature review. Certain questions arise about the interpretability, the kernel size in the trained architecture is too small. Hence, in terms of learned features, their kernel seems rather small to learn human interpretable features. One question is how the interpretability of their framework is useful.
>
> A: Interpretability in our context means mathematical interpretability. The decoder was derived to be a principled inverse of the encoder. Thus our architecture is naturally suited for autoencoding and generative tasks. Besides that, the output features of the encoder are hierarchical sparse codes, so the feature space is structured, hence interpretable. The sparse coding approach yields benefits such as the capability to perform well in denoising tasks (see section E of the Appendix).

---

> ### Author Response · Authors · 2022-11-19
> **Response to Q6~Q10 of Reviewer Ti8A**
>
>
> > Q6. The paper confuses two concepts of "image generation" and "image reconstruction". In many of the examples, for example, Figure 2, the authors are visualizing the network reconstruction (pass the image into the encoders, and then decoder). This does not image generation; the code is created by passing an image into the encoder.
>
> A: Thank you for pointing it out. We use an explicit generative model for natural images to formulate our network, and the network certainly has mechanisms to generate images. However, we mainly focus on image reconstruction in this work. We have made the appropriate wording changes in the revised version.
>
> > Q7. In Section 4.2, the authors say that "... reconstructing the samples with representation closest to these principal components". Where are the samples coming from? From my understanding, there seems to be a set of encodings from a set of images. The method is using those encodings that are closest to the principal components of the representations to reconstruct an image. Again, there is no sampling as in GANs. This is not an image generation, and looking into Principal Components of a set of representations can be applied to other autoencoders. Please elaborate.
>
> A: Thank you for pointing it out. We made the claim clear in the revised version. We also wish to point out that unless the autoencoder has a linear discriminative feature space (for example with features distributed as sparse codes or lying on orthogonal subspaces) one cannot reliably use principal components in the representation space to generate data from the original distribution.
>
> > Q8. What is the significance of large datasets that the author emphasizes if the goal is not classification, but image reconstruction? At the end of the day, all networks are being trained by gradient descent (hence not all data is going to be used at once for parameter updates). I do find the contribution of the fact that they are the first to train on imagenet-1k for image reconstruction (not classification) to be a minor contribution.
>
> A: Many works have tried to implement sparse coding-based methods for large datasets before (see  [Aberdam et al. (2020)]). However, they all failed to varying degrees; we are the first to succeed and achieve performance near the state-of-the-art.  The challenges for the previous work to scale up to ImageNet are hyperparameters like step size and the number of iterations for FISTA, normalization of the dictionary, convolution version FISTA, et al.
>
>
> > Q9. I recommend adding SOTA to Table 1, to give the reader a perspective on how far the performance is from SOTA. Perhaps to help you with your arguments on scaling up csc but not comparing to SOTA, I suggest including an extra column to report on network parameters and computational efficiency.
>
> A: Thank you for the comments. Due to the page limit, we added Table 12 in the appendix; Table 12 has all information from Table 1, plus the performance of some SOTA models, as well as network size and training time.
>
>
> > Q10. The paper frequently makes statements that are not precise and need elaboration. For example, prior to Section 3, it states "... have several good benefits unknown to any of the previous generative methods". Please be precise and explain the benefit and specify prior works.
>
> A: Thank you for the comments. Here the benefits mean the generalizability (section 4.3) to unseen datasets and the stability of training (section 4.4). We made this claim clear in the revised version.

---

> ### Author Response · Authors · 2022-11-25
> **Further Discussion with the Reviewer**
>
> Dear Reviewer Ti8A
>
> We thank you for the precious review time and valuable comments. We have provided corresponding responses and results, which we believe have covered your concerns. We hope to further discuss with you whether or not your concerns have been addressed. Please let us know if you still have any unclear parts of our work.

---

> > ### Comment · Reviewer_Ti8A · 2022-11-28
> > **Reviewer comments after reading authors' responses.**
> >
> > I thank the authors for their responses and time in providing additional information. Here are some of my concerns that are not fully addressed.
> >
> > - I thank the authors for adding "autoencoding" to resolve the confusion regarding "generation". However, the confusion is not fully resolved. There is rich literature on using sparse coding or other generative models to construct a deep neural network. Although the resulting networks are based on generative models, they are of form autoencoders: they map data into a representation and then decode the representation (sparse code) to reconstruct data. Hence, it is misleading to refer to such networks as generative networks. Following this, the scalability limitations of sparse coding-based generative networks pointed out by this paper (citing Aberdam et al. (2020)) may be valid only for generative networks. The scalability of deep networks based on a sparse coding generative model which is the case of this paper is known in the literature for image denoising (see [2,3] in my earlier comment). Given the autoencoding nature of the framework in this paper, their model is comparable to the sparse-coding-based denoising networks where the input noise is 0.
> >
> > - The additional denoising experiment needs improvement. Please report noisy PSNR. Papers that are focused on denoising task such as ([2] in my earlier comments) and DnCNNs uses a standard dataset of BSD68 to report test results and baselines which are tuned and published in the literature. Such comparison is recommended for a fair denoising comparison.
> >
> > - Following on the authors' comments on the unreliable usage of PC in the representation space of a generic network to generative image (data), it would be nice to see the outperformance of their sparse-coding-based model compared to the case where PC fails. Moreover, my concern regarding the lack of a baseline for figures of image generation quality is not fully addressed. In the ablation study, it is hard to evaluate the performance. For example, the reconstruction of CIFAR100 by Conv-AE is visually similar to CSC-AE; it seems Conv-AE is able to generalize similarly to CSC-AE to unseen data. Please clarify the conclusion and perhaps, add a quantifiable comparison.
> >
> > - The addition of information on the efficiency of the network in training and its model size against the SOTA baselines is an advantage that is added to the paper. However, with the claim of the paper to scale CSC to ImageNet, the Table still lacks a comparison with the baseline on this dataset.
> >
> > - (minor) Similar to comments by another reviewer. The filters (dictionaries) do not look interpretable (edge-detectors) from the perspective of a sparse coding model trained on natural images.

---

> > > ### Author Response · Authors · 2022-12-07
> > > **Further discussion with Reviewer Ti8A (Part 2/2)**
> > >
> > > > Q3. Following on the authors' comments on the unreliable usage of PC in the representation space of a generic network to generative image (data), it would be nice to see the outperformance of their sparse-coding-based model compared to the case where PC fails. Moreover, my concern regarding the lack of a baseline for figures of image generation quality is not fully addressed. In the ablation study, it is hard to evaluate the performance. For example, the reconstruction of CIFAR100 by Conv-AE is visually similar to CSC-AE; it seems Conv-AE is able to generalize similarly to CSC-AE to unseen data. Please clarify the conclusion and perhaps, add a quantifiable comparison.
> > >
> > > A:Thank you for the comments! We think is a really good question and want to answer in two aspects:
> > > Comparing to Conv-AE, our method learns more structured representation even for out-of-domain data
> > > Please refer to this anonymous link: https://alpha.glilmu.com/i/2022/12/07/p1evxe.png, the figure compares the PCA of learned features by the naive AE and CSC-CTRL. x-axis means the index of a PCA component, y-axis means the value of it. Larger singular value implies the feature space learned by CSC-CTRL spans more  space than the AE’s.  From the graph, we draw the conclusion that even for out-of-domain data, our method learns a more structured representation.
> > > Empirically, CSC-Layer gives better generalizability.
> > > The following table shows that CSC-layer brings good generalizability compared to conv layer. The second column means model size, the third and fourth column means the PSNR metric, the last column ‘Delta’ reports the difference of the PSNR of CIFAR-10 and CIFAR-100. It measures how the PSNR drops when the model transfers from CIFAR-10 to CIFAR-100. Smaller Delta implies better generalizability. Note here CSC based methods only needs half of the model size because layers in the decoder are shared learnable convolutional dictionary used in the encoder.
> > > Under slightly “unfair” experiment conditions, we observe that CSC layers give better generalizability than the Conv layer. If we compare the PSNR of CSC-AE and Conv-AE on CIFAR-10, the CSC-AE is only 1 point worse than Conv-AE even under the unfair condition with only half model size. We do not increase the model size of CSC-AE to the same as Conv-AE because we want to maintain the same network architecture.
> > >
> > >
> > > |            | Model Size | CIFAR-10 | CIFAR-100 | Delta  |
> > > |------------|------------|----------|-----------|--------|
> > > | CSC-based  |            |          |           |        |
> > > | CSC-AE     | 0.5M       | 29.213   | 29.008    | 0.205  |
> > > | CSC-CTRL   | 0.5M       | 24.506   | 23.582    | 0.924  |
> > > | Conv-based |            |          |           |        |
> > > | Conv-AE    | 1.0M       | 30.264   | 30.005    | 0.259  |
> > > | Conv-CTRL  | 1.0M       | 22.554   | 21.508    | 1.046  |
> > >
> > >
> > > > Q4. The addition of information on the efficiency of the network in training and its model size against the SOTA baselines is an advantage that is added to the paper. However, with the claim of the paper to scale CSC to ImageNet, the Table still lacks a comparison with the baseline on this dataset.
> > >
> > > A: Thank you for the comments. Here is the training time and model size information on ImageNet.
> > >
> > > |          | Model size | Train Time |
> > > |----------|------------|------------|
> > > | SNGAN    | 2.6 M      | >144 h     |
> > > | CSC-CTRL | 1.0 M      | 55 h       |
> > > We observe that our method is more “efficient” as it uses smaller networks and takes less time to train.
> > >
> > > We hope the above answers have clarified your questions. We would like to thank you again for engaging with us and trying to make this paper a better work! Please do not hesitate to contact us if you have any further questions.

---

> > > > ### Comment · Reviewer_Ti8A · 2022-12-09
> > > > **Raised some concerns**
> > > >
> > > > I thank the authors for providing additional analysis/results. I can agree with the authors that their framework in using CTRL provides more structure in the latent. However, I raise concerns regarding reporting numbers from experiments that are not thoroughly done/reported or do not show clearly the advantages of CSC-CTRL.
> > > >
> > > > For additional experiments on the BSD68 dataset, please clarify the train dataset and test for all the methods (the table says BSD68, but the text says test on CBSD68). The authors' explanation implies that CSC-CTRL has only been seen (trained by clean images); please report if the same process is followed for the baselines. Please report if the DnCNNs reported number is taken from published works or if the authors have trained the network themselves. If the latter, please provide more details (the size of extracted patches, depth of the network, number of filters, a comparison on #trainable parameters, etc.).
> > > >
> > > > I do not find the clustering experiment thorough. It's not clear why [3] results in worse clustering performance than the raw. Sparse coding has been shown to perform better than raw pixel clustering in traditional optimization settings. The framework in [3] has a variant to strictly follow a sparse coding model. I briefly looked into the framework in [3]; they seem to propose variants of their approach. Please report which one is used for the clustering task. More information (convolutional or dense model, kernel size, number of kernels, depth of encoders, how the sparsifying regularizer is tuned, and any processing on the code before clustering) is needed. For example, in using a sparse coding model for clustering, the regularization parameter enforcing sparsity can play a crucial role.
> > > >
> > > > This table concludes that CSC provides better generalization from CIFAR10 to CIFAR100. However, 1) compared to generic AE, CTRL results in an increase in Delta, and 2) the addition of CTRL results in much lower PSNR compared to AE. This table shows the disadvantages of CTRL. Can the authors clarify this conclusion, if that's not the case?
> > > >
> > > > I recommend the authors add the last table (on model size and train time) to the paper.

---

> > > > > ### Author Response · Authors · 2022-12-10
> > > > > **Further discussion with the Reviewer & Thank you for engaging with us**
> > > > >
> > > > > Thank you for the prompt response and constructive suggestions! It has been a great pleasure having you as our reviewer! For the additional comments, we have prepared some clarifications hoping to address your concerns.
> > > > > >Q1 For additional experiments on the BSD68 dataset, please clarify the train dataset and test for all the methods (the table says BSD68, but the text says test on CBSD68). The authors' explanation implies that CSC-CTRL has only been seen (trained by clean images); please report if the same process is followed for the baselines. Please report the training details of DnCNNs
> > > > >
> > > > > A:  Sorry for the confusion! We conducted our experiments on CBSD68. There was a typo on the table. We adopt the result of DnCNNs from https://github.com/mogvision/ADL. Due to limited time in rebuttal, we did not change the implementation of DnCNN (baseline). For the baseline DnCNN, it trains a network which takes the noised image as input, and then gets the noise as output. It trained the network in a supervised way. DnCNN has access to noisy images during the training stage. We consider it as an advantage of our method, as our method does not require access to noised data.
> > > > >
> > > > >
> > > > > >Q2 The framework in [3] has a variant to strictly follow a sparse coding model. I briefly looked into the framework in [3]; they seem to propose variants of their approach. Please report which one is used for the clustering task. More information (convolutional or dense model, kernel size, number of kernels, depth of encoders, how the sparsifying regularizer is tuned, and any processing on the code before clustering) is needed. For example, in using a sparse coding model for clustering, the regularization parameter enforcing sparsity can play a crucial role.
> > > > >
> > > > > A:  We chose the DCEA constrained (DCEA-C) one with Gaussian observations from [3]. We choose the convolutional version, the kernel size is 4x4, 64 channels, and 15 unfolded encoder iterations.. For optimization strategy, we follow the setting from [3], the ADAM optimizer with an initial learning rate 10^-3, which decreases by a factor of 0.8 every 25 epochs, and train the network for 400 epochs. We chose 0.1 lambda (coefficient of sparsifying regularizers), it will be fixed during the training. Same as we mentioned before, we just regard the DCEA-C as a feature extractor and then classify by a KNN classifier. Following your suggestion, we did an ablation study on the value of sparsifying regularizers(lambda) to see its impact on the performance of classification. From the table below, we observe that increasing the L1 regularization will slightly improve the classification accuracy but stops at lambda=0.2 and decreases when it increases to 0.5. These values are still around the value attained by the raw images, consistent with our claim that our method learns a more structured representation than methods like [3].
> > > > >
> > > > > | lambda | 0.01 | 0.05 | 0.1  |  0.2 | 0.5  |
> > > > > |--------|------|------|------|------|------|
> > > > > |[3]|30.12%|30.69%|31.72%|34.17%|32.91%|
> > > > >
> > > > >
> > > > > > Q3. This table concludes that CSC provides better generalization from CIFAR10 to CIFAR100. However, 1) compared to generic AE, CTRL results in an increase in Delta, and 2) the addition of CTRL results in much lower PSNR compared to AE. This table shows the disadvantages of CTRL. Can the authors clarify this conclusion, if that's not the case?
> > > > >
> > > > >
> > > > > A: Thank you for raising this concern! We think our work is composed of two parts: convolutional sparse coding(CSC) layers and CTRL. The purpose of this table is to show that CSC layers encourage better transferability to out-of-domain datasets.  Additionally, we observe from the table that CTRL-based methods (conv-CTRL and csc-CTRL) have a lower PSNR value. It is expected because PSNR measures the details in images, which can be encouraged by the MSE loss. While in the CTRL formulation, there is no explicit term to promote this pixel-wise consistency (such as MSE in the Autoencoders.) We think it is the precise value of our work to show that CSC networks plus CTRL can result in satisfying autoencoding while no explicit MSE loss is used. Additionally, we would like to highlight that the PSNR is not the only metric to measure the performance of an autoencoding framework. The value of this work is showing that autoencoding can be achieved without explicit MSE loss with CSC-layer + CTRL and enjoys many additional benefits such as better classification accuracy (shown below), more structured representations (in the previous replies) and better image denoising (in Appendix E.3 Table.9 ).
> > > > > For the additional table, we will add it to the paper when we can change the paper. Thank you for the value suggestion!
> > > > >
> > > > > | KNN-classifier|Conv-AE | CSC-CTRL |
> > > > > |-------------------|---------- |----------|
> > > > > | CIFAR-10 test set |34.76%  | 48.96%   |
> > > > >
> > > > > Overall, we thank the reviewer for sticking with us and trying to make this work a better one!

---

> > > > > > ### Comment · Reviewer_Ti8A · 2022-12-11
> > > > > > **Answer to Authors' Response**
> > > > > >
> > > > > > I thank the authors for providing clarifications and additional information in this short period of time. I very much appreciate the authors' effort specifically in showing that CSC-CTRL has a more structured representation than CSC-AE. However, some concerns still remain which I explain below. I hope this is helpful. I keep my score.
> > > > > >
> > > > > > - It is surprising (nice) that the proposed method can denoise images unsupervised (without the knowledge of the noise distribution). Additional thought: Is the method successful in denoising other noise levels or other noise distributions? I recommend the authors, if making a statement about denoising in the paper, leave some room for discussion. Here is a concern:  in this denoising experiment, CSC-CTLR performs very well in terms of PSNR. However, CSC-CTLR has much lower PSNR compared to CSC-AE on in-domain data in the out-of-domain experiment. This needs explanation (there seems to be a contradiction).
> > > > > >
> > > > > > - I still have concerns about the clustering experiment. For comparison to being fair, additional information on the number of features used in the proposed method is needed. Given this paper [@] from 2011 on unsupervised features learning on CIFAR10, the number of features, and kernel size have a significant effect on the final clustering performance. For example, an increasing number of features from 100 to 1000 can improve performance by 10% (this paper does classification on unsupervised extracted features) [@]. Moreover, the kernel size of 4x4 and 64 features seem rather small which is used on the baseline DCEA-C. Could authors report the number of features used by CSC-CTRL in clustering? My comments above should be addressed for this comparison to be included in the paper.
> > > > > >
> > > > > > - This last table on comparison CSC-AE and CSC-CTRL is a fair comparison and recommended to be included. I am happy with this table showing that CTRL provides more structure. However, this structure does not seem to help to reconstruct images, as shown by authors in answer to my Q3 previously (this is an important concern as the goal of the paper is to provide a method for image generation/autoencoding).
> > > > > >
> > > > > >
> > > > > > [@] Adam Coates, Andrew Ng, Honglak Lee Proceedings of the Fourteenth International Conference on Artificial Intelligence and Statistics, PMLR 15:215-223, 2011.

---

> > > > > > > ### Author Response · Authors · 2022-12-12
> > > > > > > **Response to the Reviewer and Thank you so much for the discussion**
> > > > > > >
> > > > > > > Thank you so much for the prompt reply! We think your questions are very good and highly valuable. We have provided some clarifications for your concerns and hope to address them
> > > > > > >
> > > > > > > >Q1 It is surprising (nice) that the proposed method can denoise images unsupervised (without the knowledge of the noise distribution). Additional thought: Is the method successful in denoising other noise levels or other noise distributions? I recommend the authors, if making a statement about denoising in the paper, leave some room for discussion. Here is a concern: in this denoising experiment, CSC-CTLR performs very well in terms of PSNR. However, CSC-CTLR has much lower PSNR compared to CSC-AE on in-domain data in the out-of-domain experiment. This needs explanation (there seems to be a contradiction).
> > > > > > >
> > > > > > > A:  Thank you for raising this point!  For denoising other noise levels or other noise distributions, we leave it for future studies due to limited time in the rebuttal period. For your concern, we’ll provide a short explanation below. The gooding denoising effect (as shown in PSNR) is credited to our CSC layer and CTRL formulation for finding a meaningful structured representation. So, when the input noise to perturbed, we can recover the structure in images.
> > > > > > >
> > > > > > >
> > > > > > >
> > > > > > > > Q2 I still have concerns about the clustering experiment. For comparison to being fair, additional information on the number of features used in the proposed method is needed. Given this paper [@] from 2011 on unsupervised features learning on CIFAR10, the number of features, and kernel size have a significant effect on the final clustering performance. For example, an increasing number of features from 100 to 1000 can improve performance by 10% (this paper does classification on unsupervised extracted features) [@]. Moreover, the kernel size of 4x4 and 64 features seem rather small which is used on the baseline DCEA-C. Could authors report the number of features used by CSC-CTRL in clustering? My comments above should be addressed for this comparison to be included in the paper.
> > > > > > >
> > > > > > > A: Thank you for your comments! Since we implemented the DCEA-C in convolutional version with stride=2 and 64 output channels, the dimension of output sparse code is 64x16x16=16384 (16x16 means the size of one feature map down sampled by stride 2 convolution.)  which is much larger than the dimension (512) we used in CSC-CTRL. We acknowledge your concern that a 16384 dimension is not the best setting of DCEA-C, we conduct further ablation study on the number of feature maps from 64x16x16 to 512x16x16, and show the classification accuracy in the table. We observe that DCEA-C reaches its top performance around 256x16x16, which is still much lower than our performance (48.96%), despite we have used a much smaller dimension(512).
> > > > > > >
> > > > > > > | #features | 64x  | 128x | 256x | 512x |
> > > > > > > |-----------|------|------|------|------|
> > > > > > > | [3]       |34.17 | 36.78| 37.13| 36.89|
> > > > > > >
> > > > > > > > Q3 This last table on comparison CSC-AE and CSC-CTRL is a fair comparison and recommended to be included. I am happy with this table showing that CTRL provides more structure. However, this structure does not seem to help to reconstruct images, as shown by authors in answer to my Q3 previously (this is an important concern as the goal of the paper is to provide a method for image generation/autoencoding).
> > > > > > >
> > > > > > > A:  We agree that this is an important point that needs to be discussed. The purpose of this work is to propose a new method that attains meaningful structures and satisfying autoencoding.  In fact, structured representation and good autoencoding are not correlated.  One can see the autoencoding in  Figure.1 of AASAE[1] as an example (structured representation leads to very blurry autoencoding). It is the value of our work to seek an optimal balance between meaningful structures and good autoencoding. It enables our method to identify structures in images by simply calculating the principal components (Figure 4) while enjoying good reconstruction. And, we would like to highlight, despite lower PSNR, we observe from images (Figure 2, 3, 8, and 9) that, visually, the autoencoding is very convincingly good. We think it is a valuable contribution to the community since our method does not explicitly use MSE or other loss functions to directly enforce sample-wise similarity.
> > > > > > >
> > > > > > > > Thank you for all these discussions!
> > > > > > >
> > > > > > > For all of the discussions we have, we’ll add them to the paper when we have the opportunity to do so. In case we don’t have a chance to update our answer because the rebuttal stage is approaching its end, we just want to say that it has been a great pleasure to discuss our work with you! We think the discussion with you is what makes the extended rebuttal period in ICLR so valuable. Thank you for engaging with us!
> > > > > > >
> > > > > > > [1]: Falcon, William, et al. "AASAE: Augmentation-Augmented Variational Autoencoders." arXiv preprint arXiv:2107.12329 (2021).

---

> > > > > > > > ### Comment · Reviewer_Ti8A · 2022-12-13
> > > > > > > > **Continue discussion**
> > > > > > > >
> > > > > > > > I thank the authors for addressing my comments. My Q2 on the clustering experiment is addressed.
> > > > > > > >
> > > > > > > > I do understand that CTRL provides certain structures in the representation and it should be appreciated. However, I still do not understand how CSC-CTRL is able to perform better than DnCNNs for denoising and reconstructing well in terms of PSNR (from Q1) but fails to have better PSNR in Q3 compared to CSC-AE. Could the authors please explain/justify this behavior?
> > > > > > > >
> > > > > > > > Regarding image quality, I recommend the authors add the SSIM metric along with PSNR which is more representative of the perceptual quality of images.

---

> > > > > > > > > ### Author Response · Authors · 2022-12-13
> > > > > > > > > **Response to Continual discussion**
> > > > > > > > >
> > > > > > > > >
> > > > > > > > > We thank the reviewer for engaging with us until the last moment in the rebuttal period! We have really enjoyed the discussion we had with you.
> > > > > > > > > >Q I do understand that CTRL provides certain structures in the representation and it should be appreciated. However, I still do not understand how CSC-CTRL is able to perform better than DnCNNs for denoising and reconstructing well in terms of PSNR (from Q1) but fails to have better PSNR in Q3 compared to CSC-AE. Could the authors please explain/justify this behavior?
> > > > > > > > >
> > > > > > > > > A: Hi, thank you for pointing it out! For the part related to denoising, we are in fact inspired by [3] you raised and increased the number of iterations in the FISTA algorithm for better denoising. We have provided a table that investigates the influence of FISTA iteration on the performance of PSNR. In all other experiments, we stick to the default # of Fista iteration, which is 2, and under performed in the PSNR on CIFAR-10. For denoising, since it is a more complicated task, we increase the # of Fista Iteration similar to [3] and got the results in the table. We apologize for the confusion here. Be that as it may, we think this shows great promise of our method in adapting to different tasks. We'll make sure to add a section to discuss this point in the revised version later.
> > > > > > > > >
> > > > > > > > > | # of FISTA iteration             |2          | 8        | 16       |
> > > > > > > > > |---------------------------       |---------- |----------| ---------|
> > > > > > > > > | CIFAR-10 clean input             |24.506     |29.942    |32.510    |
> > > > > > > > > | CIFAR-10 noise input (delta=0.1) |22.891     |28.315    |31.679    |
> > > > > > > > >
> > > > > > > > >
> > > > > > > > > > Q Regarding image quality, I recommend the authors add the SSIM metric along with PSNR which is more representative of the perceptual quality of images.
> > > > > > > > >
> > > > > > > > > A: Thank you for the suggestion! Due to limited time in the rebuttal period, we will add them to our paper when we can change the paper. We agree that it is a more representative measure!
> > > > > > > > >
> > > > > > > > > Since this is very likely to be our last reply because the discussion phase is ending very soon. We would like to thank you again for engagements!

---

> > > > > > > > > > ### Comment · Reviewer_Ti8A · 2022-12-14
> > > > > > > > > > **Can the authors show that CSC-CTRL is better than CSC-AE or CTRL. The current result does not show that CSC-CTRL is better than CSC-AE for generation/reconstruction.**
> > > > > > > > > >
> > > > > > > > > > I thank the authors for the explanation. Now the differences in performance are clear.
> > > > > > > > > >
> > > > > > > > > > Using a low number of CSC FISTA iterations (which implicitly is aiming to provide better reconstruction) can result in very low PSNR in CSC-CTRL which does not have a reconstruction loss in the image space during training. The authors have increased the number of FISTA iterations in their denoising performance to achieve high PSNR, which is a measure of reconstruction in image space. For out-of-domain experiments (CIFAR10 to CIFAR100), the authors have seemed to use low FISTA iterations, and this explains the low PSNR performance of CTRL compared to AE. In my opinion, to be able to show/evaluate the usefulness of the structured representation CTRL when combine with CSC, the network should be unfolded sufficiently. In this current version of the out-of-domain experiment, CSC-CTRL has clear disadvantages over CSC-AE in both in and out-of-domain performance. I very much appreciate the authors' effort in addressing my comments. Focusing on the presented result, below can increase my score:
> > > > > > > > > >
> > > > > > > > > > If the authors show that CSC-CTRL has a close and better performance to CSC-AE in terms of SSIM/PSNR/etc. for image generation, and out-of-domain and in-domain reconstruction. My focus on showing the advantages of CTRL is because CTRL is a newer concept, and certain advantages of CSC have already been studied and are known to the machine learning community (my priors citations).
> > > > > > > > > >
> > > > > > > > > > Looking again into the paper, in comparisons (e.g., Table 1, Figure 7), only CTRL and CSC-CTRL are included. CSC-AE is missing. To highlight the benefit of your method, you may want to show that you are better than CTRL and CSC-AE.

---

> > > > > > > > > > > ### Author Response · Authors · 2022-12-14
> > > > > > > > > > > **Reply to Reviewer's Comment**
> > > > > > > > > > >
> > > > > > > > > > > We thank the reviewer for the concrete suggestions! For your comments, we have prepared some additional results hoping to address your concern.
> > > > > > > > > > > >Q Focusing on the presented result, below can increase my score:
> > > > > > > > > > > If the authors show that CSC-CTRL has a close and better performance to CSC-AE in terms of SSIM/PSNR/etc. for image generation, and out-of-domain and in-domain reconstruction. My focus on showing the advantages of CTRL is because CTRL is a newer concept, and certain advantages of CSC have already been studied and are known to the machine learning community (my priors citations).
> > > > > > > > > > > Looking again into the paper, in comparisons (e.g., Table 1, Figure 7), only CTRL and CSC-CTRL are included. CSC-AE is missing. To highlight the benefit of your method, you may want to show that you are better than CTRL and CSC-AE.
> > > > > > > > > > >
> > > > > > > > > > > A: Thank you for this comment! We agree with your point and present the results of our CSC-CTRL (with 16 FISTA Iteration) below.
> > > > > > > > > > >
> > > > > > > > > > > |method/ (PSNR)                    |CSC-AE     | CSC-CTRL(FISTA iteration=16)       |
> > > > > > > > > > > |---------------------------       |---------- |------------------------------------|
> > > > > > > > > > > | CIFAR-10 test set                |29.213     |**32.510**                              |
> > > > > > > > > > > | CIFAR-100 test set               |29.008     |**31.599**                              |
> > > > > > > > > > >
> > > > > > > > > > > Due to limited time in the rebuttal period, we show the PSNR number right now and we will add SSIM in the revised version. From the table, we see that with 16 FISTA Iterations, CSC-CTRL outperforms CSC-AE in both in domain(CIFAR-10) and out of domain (CIFAR-100) measured by PSNR.
> > > > > > > > > > >
> > > > > > > > > > > We agree that CSC-AE should also be added to the table. So we will make sure to include it in the revised version.
> > > > > > > > > > > Thank you for your suggestion! You have really helped us make this piece a better work. It was fortunate for us to have you as our reviewer!

---

> > > > > > > > > > > > ### Comment · Reviewer_Ti8A · 2022-12-14
> > > > > > > > > > > > **Does CSC-AE also have 16 FISTA iteration?**
> > > > > > > > > > > >
> > > > > > > > > > > > I thank the authors for this fast response and the additional reported numbers. Could the authors clarify if CSC-AE has similar FISTA iterations as CSC-CTRL? If not, could you report CSC-AE with 16 FISTA iterations for a fair comparison?

---

> > > > > > > > > > > > > ### Author Response · Authors · 2022-12-14
> > > > > > > > > > > > > **Response to Fista Iteration**
> > > > > > > > > > > > >
> > > > > > > > > > > > > > I thank the authors for this fast response and the additional reported numbers. Could the authors clarify if CSC-AE has similar FISTA iterations as CSC-CTRL? If not, could you report CSC-AE with 16 FISTA iterations for a fair comparison?
> > > > > > > > > > > > >
> > > > > > > > > > > > > Thank you for the quick reply! It is an interesting point to discuss. Visually, we found that FISTA iterations have little impact on CSC-AE. So we reported the CSC-AE with 2 FISTA iterations in the previous table. We think MSE loss in CSC-AE is already very strong to enforce sample-wise consistency.
> > > > > > > > > > > > > Here, following your advice, we measure the results of CSC-AE(Fista Iteration=16) and present the results below.
> > > > > > > > > > > > >
> > > > > > > > > > > > > |method/ (PSNR)                |CSC-AE（FISTA iteration=16）| CSC-CTRL(FISTA iteration=16)       |
> > > > > > > > > > > > > |---------------------------   |----------------------- |----------------------------------------|
> > > > > > > > > > > > > | CIFAR-10 test set            |31.926                  |32.510                    |
> > > > > > > > > > > > > | CIFAR-100 test set           |31.031                  |31.599                    |
> > > > > > > > > > > > >
> > > > > > > > > > > > > From the table, we observe that CSC-AE and CSC-CTRL achieve very similar results, with CSC-CTRL slightly better than CSC-AE.
> > > > > > > > > > > > >
> > > > > > > > > > > > > Thank you for the quick response!

---

> > > > > > > > > > > > > > ### Comment · Reviewer_Ti8A · 2022-12-14
> > > > > > > > > > > > > > **Final comment - score**
> > > > > > > > > > > > > >
> > > > > > > > > > > > > > I thank the authors for the additional information. My conclusion from the latest analysis is that the CTRL structure helps reconstruction/generation. I recommend the authors clearly state the number of unfolded FISTA iterations for any result they report.
> > > > > > > > > > > > > >
> > > > > > > > > > > > > > I have increased my score. However, I cannot strongly support the work as 1) the novelty arguments stay there and 2) it's not clear how CSC-AE performs against the baselines in the main experiments (Table 1, Figure 7, ...), and I won't be able to see/evaluate such modifications (as authors cannot change the paper in this post-rebuttal period).
> > > > > > > > > > > > > >
> > > > > > > > > > > > > > I hope the discussion was overall helpful to the authors to improve the paper.

---

> > > ### Author Response · Authors · 2022-12-07
> > > **Further discussion with Reviewer Ti8A (Part 1/2)**
> > >
> > > We would like to thank the reviewer for the valuable comments! We have prepared some answers hoping to clarify your question. Because there were some additional experiments raised, we spent some time running them and replied a little late. We appreciate the reviewer for engaging with us!
> > >
> > > > Q1. The additional denoising experiment needs improvement. Please report noisy PSNR. Papers that are focused on denoising task such as ([2] in my earlier comments) and DnCNNs uses a standard dataset of BSD68 to report test results and baselines which are tuned and published in the literature. Such comparison is recommended for a fair denoising comparison.
> > >
> > > A: We thank the reviewer for raising this question. The following table shows the comparison of CSC-CTRL with other three denoising models on CIFAR10 with noise level delta=0.1 and delta=0.5. In most denoise settings, they directly add noise to RGB value (0-255), and then normalize the image to 0-1 value scale. But in our experiment, we add noise after image normalization. Hence, the delta=0.1 in our setting equals delta=25.5 in their setting.
> > >
> > > | Noise level (delta=0.5) | PSNR   |
> > > |-------------------------|--------|
> > > | noise input             | 12.050 |
> > > | CSC-CTRL                | **17.094** |
> > > | AE                      | 13.141 |
> > > | AE w Sym                | 15.784 |
> > > | DnCNNs                  | 16.142 |
> > >
> > > | Noise level (delta=0.1) | PSNR   |
> > > |-------------------------|--------|
> > > | noise input             | 26.027 |
> > > | CSC-CTRL                | **31.679** |
> > > | AE                      | 24.830 |
> > > | AE w Sym                | 28.254 |
> > > | DnCNNs                  | 28.992 |
> > >
> > > Empirically, we observe that our method has a better denoising effect even though our method is not designed for it! For the CBSD68 dataset, the following table compares the PSNR of CSC-CTRL and DnCNNs on CBSD68 (delta=50). The CSC-CTRL is only trained on the original clean BSD68 images and then denoise on gaussian noise with 50 std. The metric  PSNR of CSC-CTRL is better than DnCNNs.
> > >
> > > | BSD68 (delta=50)        | PSNR   |
> > > |--------------------------|--------|
> > > | noise input              | 18.418 |
> > > | CSC-CTRL                 | **29.78**  |
> > > | DnCNNs                   | 28.16  |
> > >
> > >
> > > > Q2. I thank the authors for adding "autoencoding" to resolve the confusion regarding "generation". However, the confusion is not fully resolved. There is rich literature on using sparse coding or other generative models to construct a deep neural network. Although the resulting networks are based on generative models, they are of form autoencoders: they map data into a representation and then decode the representation (sparse code) to reconstruct data. Hence, it is misleading to refer to such networks as generative networks. Following this, the scalability limitations of sparse coding-based generative networks pointed out by this paper (citing Aberdam et al. (2020)) may be valid only for generative networks. The scalability of deep networks based on a sparse coding generative model which is the case of this paper is known in the literature for image denoising (see [2,3] in my earlier comment). Given the autoencoding nature of the framework in this paper, their model is comparable to the sparse-coding-based denoising networks where the input noise is 0.
> > >
> > > A: The methods in [2, 3] are the equivalent to one layer version of our CSC-AE. The purpose of introducing one CSC layer is image denoising. Hence, the learned sparse code of their method is less structured. Our CSC-CTRL model does autoencoding through a multi-CSC-layers network with the rate reduction objective and closed loop framework. The learned feature contains more structure. We verify it using a knn-classifier on the learned features.
> > > Within this short rebuttal period, we reimplement the method from [3] and compare it with our CSC-CTRL. Both methods are trained on CIFAR-10 training set, and test the accuracy on CIFAR-10 testing set through KNN. The results could be found in the following table.
> > >
> > > | KNN-classifier    | [3]    | Raw CIFAR-10 | CSC-CTRL |
> > > |-------------------|--------|----------    |----------|
> > > | CIFAR-10 test set | 31.72% |33.99%        | 48.96%   |
> > >
> > > As we see, the method from [3] gets 31.72% testing accuracy where the test accuracy of KNN on the original image is already 33.99%. It verifies the claim that sparse code learned from denoising [3] does not contain much structure. It only serves the purpose of image denoising; Whereas our method aims to learn a more structured representation.
> > > We think this result distinguishes our method from [2,3]. And it is precisely the value of our work to extend [2,3] to multi-layer networks and more complicated dataset like CIFAR-100, STL-10, ImageNet-1k(64*64)

---

### Official Review · Reviewer_uge7 · 2022-10-25

**Confidence:** 4
**Correctness:** 3
**Technical Novelty And Significance:** 3
**Empirical Novelty And Significance:** 3
**Recommendation:** 6

**Clarity, Quality, Novelty And Reproducibility:**

The paper is written clearly. The idea of combining convolutional sparse coding with closed loop transcription is novel to the best of my knowledge. The implementation details are provided in the appendix. Having the implementation code would help reproduce the results presented in the work.


**Strength And Weaknesses:**

**Strengths**

- The proposed architecture combining convolutional sparse coding with closed loop transcription is novel.

- The proposed model is compared to and in many cases performs better than existing generative models of similar capacity on popular evaluation metrics.

- Qualitative analysis shows that images generated by the CSC-CTRL model have high visual fidelity, both sample-wise and distribution-wise.

- The paper is generally written clearly.

**Room for improvement**

- *ImageNet scale*: I believe the statement in the abstract that the CSC-CTRL model “scales up to ImageNet” might be misleading since Appendix A.2. states that the ImageNet images are resized from 224x224 to 64x64. Would the authors please clarify what ImageNet image size is used during training?

- *Statistical significance*: It would strengthen the paper’s argument to include confidence intervals in Table 1.

- *Generalization to Unseen Datasets* and *Stability of CSC-CTRL*: It would complement the qualitative analysis in figures 5 and 7 to provide quantitative analysis such as average PSNR the model achieves on: CIFAR-10 (which it is trained on) versus PSNR on CIFAR-100 in Fig 5; the average PSNR of reconstructions of the original versus noisy inputs for both the CTRL and CSC-CTRL models trained on CIFAR-10 and STL-10 (evaluated on a held-out test set).

- *Clarification on equations (9) and (10)*: In my understanding $\theta$ and $\eta$ (the encoder and decoder’s parameters) are the same since they are determined by the weights of the shared convolutional dictionaries. Is my understanding correct?

- *Related literature*: I believe it would be helpful to include reference to either [1, 2] in the main text.

- *Visualization of convolutional filters*: It would be informative to display the learned convolutional filters which is a commonly adopted practice in the related literature.

- *Reproducibility*: It would be helpful if the authors release their implementation for reproducibility.

[1] Zeiler, M.D., Krishnan, D., Taylor, G.W. and Fergus, R., 2010, June. Deconvolutional networks. In 2010 IEEE Computer Society Conference on computer vision and pattern recognition (pp. 2528-2535). IEEE.

[2] Zeiler, M.D., Taylor, G.W. and Fergus, R., 2011, November. Adaptive deconvolutional networks for mid and high level feature learning. In 2011 international conference on computer vision (pp. 2018-2025). IEEE.


**Summary Of The Paper:**

This paper proposes a model for image generation which combines convolutional sparse coding (CSC) with the closed-loop transcription (CTRL) framework by Dai et al (2022). The encoder and decoder of this CSC-CTRL model share dictionaries of convolutional kernels at each sparse coding layer. The encoder produces a sparse code at each CSC-layer by unrolling the FISTA algorithm. The decoder applies deconvolution to the final output of the encoder and each successive CSC-layer output to reconstruct the original input. The system is trained with a rate reduction objective which aims to minimize the distance between the distributions of codes produced from the original inputs and those produced from the reconstructed inputs. The proposed model is trained on CIFAR-10, STL-10, and ImageNet-1K, and achieves comparable or favorable performance when evaluated against GAN, VAE, Flow, and CTRL methods on metrics such as Inception Score and FID. Additional analysis suggests that the feature space of the proposed model preserves class information even though it is trained with an unsupervised objective. Further experimental results suggest that the CSC-CTRL model generalizes to data from classes not seen during training, is robust to noise, and is more stable than the CTRL model when trained with different batch sizes.


**Summary Of The Review:**

Overall, this work presents an interesting idea of combining convolutional sparse coding with closed loop transcription for image generation. The model is compared to generative models of similar or higher capacity and performs favorably in many cases. The qualitative results on the structure of the feature space, generalization, as well as robustness to noise are insightful. I pose some recommendations on ways to strengthen the statements in the paper by including clarifications and more analysis.

---

> ### Author Response · Authors · 2022-11-19
> **Response to Reviewer uge7**
>
>
> We thank the reviewer for their insightful comments.
>
> > Q1. ImageNet scale: I believe the statement in the abstract that the CSC-CTRL model “scales up to ImageNet” might be misleading since Appendix A.2. states that the ImageNet images are resized from 224x224 to 64x64. Would the authors please clarify what ImageNet image size is used during training?
>
> A: Thank you for pointing it out. As mentioned in Appendix A.2, we use 64x64 image size. There are two components of a “large scale dataset”: a large number of classes and a high resolution. The statement “scales up to ImageNet” in the abstract means we scale our CSC-CTRL model from 10 classes (CIFAR-10) to 1000 classes (ImageNet), and from 32x32 resolution (CIFAR-10) to 64x64 resolution (ImageNet). Both of these scale-ups were previously unachieved by any other sparse coding-based generative methods [Aberdam et al. (2020)].
>
>
> > Q2. Statistical significance: It would strengthen the paper’s argument to include confidence intervals in Table 1.
>
> A: Thank you for pointing it out. We add an ablation study in Table 11 of the Appendix to test how stable our method is with respect to different random seeds.
>
> > Q3. Generalization to Unseen Datasets and Stability of CSC-CTRL: It would complement the qualitative analysis in figures 5 and 7 to provide quantitative analysis such as average PSNR the model achieves on: CIFAR-10 (which it is trained on) versus PSNR on CIFAR-100 in Fig 5; the average PSNR of reconstructions of the original versus noisy inputs for both the CTRL and CSC-CTRL models trained on CIFAR-10 and STL-10 (evaluated on a held-out test set).
>
> A: Thank you for the comments. In the revised version, we complement the comparison for the unseen dataset on both qualitative (Figures 14 and 15) in section I of the appendix. For stability to noise, we add section E.3 in the appendix to give quantitative results.
>
>
> > Q4. Clarification on equations (9) and (10): In my understanding theta and eta (the encoder and decoder’s parameters) are the same since they are determined by the weights of the shared convolutional dictionaries. Is my understanding correct?
>
> A: Yes, your understanding is correct.
>
> > Q5. Related literature: I believe it would be helpful to include reference to either [1, 2] in the main text.
>
> A: We thank you for the reviewer’s comments. We add them to the revised version.
>
> > Q6. Visualization of convolutional filters: It would be informative to display the learned convolutional filters which is a commonly adopted practice in the related literature.
>
> A: Thank you for pointing it out. In the revised version, we visualize the dictionaries of all layers in Figure 13 of the Appendix.
>
> > Q7. Reproducibility: It would be helpful if the authors release their implementation for reproducibility.
>
> A: Sure, we will release the code after the review.

---

> > ### Comment · Reviewer_uge7 · 2022-11-22
> > **Thank you for the response! Some additional thoughts on Q3 and Q6.**
> >
> > I thank the authors for the detailed response! Please find below a couple of follow-up comments on Q3 and Q6.
> >
> > Regarding Q3 - The results in Appendix section E.9 (Table 9) speak in favor of CSC-CTRL model's robustness to noise. Going back to my original suggestion, I think it would still be meaningful to compare the CSC-CTRL model's performance in terms of PSNR measured on CIFAR-10 (which it was trained on) versus CIFAR-100 to quantify the results in Figure 5.
> >
> > Regarding Q6 - The filters do not look very interpretable but this might be the case if the level of sparsity used is relatively low.

---

> > > ### Author Response · Authors · 2022-11-22
> > > **Additional response to Q3 and Q6.**
> > >
> > > We thank the reviewer’s prompt response. For Q3, the following table reports the PSNR comparison on CIFAR-10 and CIFAR-100  testing sets, where the model only trained on CIFAR-10 training set.  The PSNR of our model only drops 0.924 when we transfer it to CIFAR-100. This reflects the good generalizability of our model. The 23.582 PSNR of CIFAR-100 is the result of Figure 5 in the main body.
> > >
> > > |      | CIFAR-10 | CIFAR-100 |
> > > |------|----------|-----------|
> > > | PSNR | 24.506   | 23.582    |
> > >
> > >
> > > For Q6, we believe that the 4 by 4 kernel size is too small to see any meaningful results. Moreover, the filter might not be ideal to visualize the sparse information, as the sparsity is contained in the feature of each layer. In measuring the sparsity, we follow the common practice in the literature and calculate the ratio of non-zero entries of the feature. We calculate the output feature of the first layer on the whole testing set of CIFAR-10. The nonzero ratio is 56% which is good evidence of the sparsity of our method.

---

> > > > ### Comment · Reviewer_uge7 · 2022-11-25
> > > > **Re: Additional response to Q3 and Q6.**
> > > >
> > > > Thank you for your response!
> > > >
> > > > Q3: Thank you for providing these additional numbers on PSNR for CIFAR-10 vs CIFAR-100 where the model is only trained on CIFAR-10 training set. It is promising that the drop in PSNR on CIFAR-100 is relatively small.
> > > >
> > > > Q6: Even though I agree that the convolutional kernel size 4x4 is relatively small, I believe it should be possible to observe the emergence of orientation/edge detectors, center surrounds, etc, if the sparsity level is sufficiently high (maybe 56% sparsity is not enough). In any case, sparsity level of 56% for CSC-CTRL model seems sufficient for it to perform favorably when compared to the CTRL and other models based on the empirical results in the paper.
> > > >
> > > > Thank you again and I do not have any additional comments at the moment.

---

### Official Review · Reviewer_b6Q7 · 2022-11-03

**Confidence:** 3
**Correctness:** 2
**Technical Novelty And Significance:** 2
**Empirical Novelty And Significance:** 1
**Recommendation:** 3

**Clarity, Quality, Novelty And Reproducibility:**

The paper's clarity needs to be improved, The experiment settings are not properly explained, and the motivation is not very clear.

**Strength And Weaknesses:**

**Strength:**
This paper has many experiments.

**Weakness:**
1. Some of the arguments are overclaimed. For instance, in Introduction: "The learned autoencoder achieves striking sample-wise consistency". Not sure if the performance is really that impressive.

2. "our method scales well to large datasets": it is unclear what "scales well" mean. Does it mean the model is able to be trained on large datasets, or achieve good performance?

3. As mentioned in Section 4, "The main message we want to convey is that the convolutional sparse coding-based deep models can indeed scale up to large-scale datasets and regenerate high-quality".  Not sure if the contribution is sufficient. At the same time, the reconstructed images do not seem to be very impressive compared to many other generative models like diffusion models, and normalizing flows.

4. The experiment settings are poorly explained. The model's performance is also very limited.

5. In section 4.1, besides showing the reconstruction qualitatively in figure 2 and 3, it would be more convincing to evaluate the reconstruction qualitatively and compare it with the baselines.

6. Table 1 is very confusing. Are the IS and FID scores evaluated on samples or reconstructions? If they are evaluated on samples, then there should be more details on sampling from the proposed model. Currently, I do not see the sampling algorithm in the paper. If they are evaluated on reconstructions, then I am not sure if the comparison with the GAN, VAE, flow baselines are fair (since they are evaluated on synthetic samples).

7. The figures and their captions are confusing. For instance, for figure 4-6, it would be much better to label the x, y-axis and each block to clarify what each image is.

8. It is unclear why the performance in figure 6 is impressive: many autoencoders will also generalize to a different datasets due to some inductive bias. Especially since CIFAR100 is relatively similar to CIFAR-10.

9. The denoising experiment in figure 7 does not seem to be working.


**Summary Of The Paper:**

This work proposes to replace the encoder and decoder with standard convolutional sparse coding and decoding layers in an autoencoder. The proposed approach can be trained on high-resolution images and can be used to reconstruct images and capture interpretable representations.

**Summary Of The Review:**

Given that the paper's clarity is limited and the experiments are also not very convincing, the paper needs to be improved for acceptance.

---

> ### Author Response · Authors · 2022-11-19
> **Response to Reviewer b6Q7**
>
> We thank the reviewer for their insightful comments.
>
> > Q1.Some of the arguments are overclaimed. For instance, in the introduction: "The learned autoencoder achieves striking sample-wise consistency". Not sure if the performance is really that impressive.
>
> A:  Please refer to Figures 4, 9, and 10 of the experiment section; the results definitely demonstrate sample-wise consistency. For the reconstruction performance, our method is still a little behind GAN/VAE based SOTA methods. However, amongst the set of sparse coding-based methods, the performance of our method is the best. We’ve made some wording changes to sharpen the claims; for example, we reword “striking sample-wise consistency” to “good sample-wise consistency”.
>
> > Q2. "our method scales well to large datasets": it is unclear what "scales well" mean. Does it mean the model is able to be trained on large datasets, or achieve good performance?
>
> A: It means our model is able to be trained on large datasets, such as ImageNet, and achieves good results only a little lower than state-of-the-art under fair comparison.
>
> > Q3. As mentioned in Section 4, "The main message we want to convey is that the convolutional sparse coding-based deep models can indeed scale up to large-scale datasets and regenerate high-quality". Not sure if the contribution is sufficient. At the same time, the reconstructed images do not seem to be very impressive compared to many other generative models like diffusion models, and normalizing flows.
>
> A: Please see our response to all reviewers at the top.
>
> > Q4. The experiment settings are poorly explained. The model's performance is also very limited.
>
> A: Due to page limits, we move detailed description of the experiment settings to Appendix A.
>
> > Q5. In section 4.1, besides showing the reconstruction qualitatively in Figure 2 and 3, it would be more convincing to evaluate the reconstruction qualitatively and compare it with the baselines.
>
> A: Thank you for pointing it out. We add a comparison with the baselines in Figures 14 and 15 of the appendix in the revised version.
>
> > Q6. Table 1 is very confusing. Are the IS and FID scores evaluated on samples or reconstructions? If they are evaluated on samples, then there should be more details on sampling from the proposed model. Currently, I do not see the sampling algorithm in the paper. If they are evaluated on reconstructions, then I am not sure if the comparison with the GAN, VAE, flow baselines are fair (since they are evaluated on synthetic samples).
>
> A: The IS and FID scores were evaluated on reconstructions. Due to page limits, we add Table 12,  which has all information from Table 1 plus some more details, to the appendix. In particular, Table 12 has the reconstruction IS/FID results of VAE based methods. We also add extra columns to report the model size and training time. This will give the reader a perspective on how we scale up our method.
>
> > Q7. The figures and their captions are confusing. For instance, for Figure 4-6, it would be much better to label the x, y-axis and each block to clarify what each image is.
>
> A: Figures 4 and 6 are qualitative results demonstrating the reconstruction quality. There is no further meaning for the axes. For Figure 5, it shows five reconstructed $\hat x=g(z)$ from $z$'s which are chosen to have the closest distances to the subspace spanned by the top four principal components of the learned features, for each of five ImageNet classes (class ''rajidae'', ''goldfish'', ''chicken'', ''bird'', ''shark''). We updated it in the caption and main body.
>
> > Q8. It is unclear why the performance in figure 6 is impressive: many autoencoders will also generalize to a different datasets due to some inductive bias. Especially since CIFAR100 is relatively similar to CIFAR-10.
>
> A: We would like to remind the reviewer that CIFAR10 is very different from CIFAR100, despite their similar names. More than 95 classes of images in CIFAR100 are different from the classes in CIFAR10. (please refer to the website of CIFAR https://www.cs.toronto.edu/~kriz/cifar.html). It is well-known that GAN models cannot generate out-of-domain datasets; we add the comparison to other methods on the unseen dataset in Figures 14, 15 in the revised version.
>
> > Q9. The denoising experiment in figure 7 does not seem to be working.
>
> A: In Figure 7, we have added **50%** of Gaussian noise which is **much larger** than the Gaussian noise from other works. Even under such extreme settings, our method achieves very satisfying denoising results. In Table 8 of the revised version, our denoising results are shown to be better than even denoising-specific methods.

---

> ### Author Response · Authors · 2022-11-25
> **Further Discussion with the Reviewer**
>
> Dear Reviewer b6Q7
>
> We thank you for the precious review time and valuable comments. We have provided corresponding responses and results, which we believe have covered your concerns. We hope to further discuss with you whether or not your concerns have been addressed. Please let us know if you still have any unclear parts of our work.

---

> ### Author Response · Authors · 2022-12-11
> **Thank you for reviewing our work**
>
> Dear Reviewer b6Q7, Since the discussion phase is about to end in a few days, we wonder if our reply properly addressed the concerns. Thank you for your valuable time and helpful suggestion!

---

### Decision · Program_Chairs · 2023-01-20

**Decision:**

Reject

**Justification For Why Not Higher Score:**

The combination of two existing approaches needs stronger intuition/justification. Showing mostly reconstruction quality is not very interesting. Reviewers had concerns on the experimental setups.


**Justification For Why Not Lower Score:**

N/A

**Metareview: Summary, Strengths And Weaknesses:**

The paper proposes a convolutional sparse coding auto-encoding model, where the encoder is a convolutional sparse coding layer solving lasso (sparse coding problem) by unrolling FISTA, and the decoder is dictated by the sparse coding/dictionary learning model. Instead of the usual reconstruction loss, the authors use closed-loop transcription (CTRL) to train the network. They show the reconstruction/generation performance of the method. In summary, the paper combines the idea of convolutional sparse coding and CTRL.

strengths:

The paper was easy to read and performed ablation studies. There are reasonable visualizations and characterizations of the method.

weaknesses:

Reviewers had concerns on the paper over-claiming the advantage in generation, while the benefit was mostly shown for the reconstruction task, and the "sampling" procedure is not the same (or as natural) as compared methods. The reviewers also think the experimental setup is problematic in using very small filters which hamper the interpretability of the method, and that the mixed results on denoising and out-of-domain reconstruction. The paper combines two existing approaches (with CTRL being more recent): the sparse coding part does involve a L2 reconstruction loss, while CTRL tries to avoid measuring reconstruction error in image space.  Thus there seems to be a contradiction between the principles of the two parts, making the combination not very natural at first look. The authors may need strong theoretical motivations to reconcile them.